# Toward Calibrated Mixture-of-Experts Under Distribution Shift

**Gina Wong**[1]  **Drew Prinster**[1]  **Suchi Saria**[1]  **Rama Chellappa**[1]  **Anqi Liu**[1]

## Abstract

Calibration aligns a model's predictive uncertainty with the frequencies of its empirical outcomes and is important for understanding and trusting reported probabilities. Recent work shows that enforcing calibration at the level of individual predictors can improve ensemble accuracy and calibration, with mixture-of-experts (MoE) models showing strong empirical improvements in particular; however, the conditions under which calibration helps MoE are not well understood. In this work, we study how MoE models behave under distribution shift, focusing on how routing mechanisms interact with expert-level calibration. We show that expert calibration is sufficient to ensure calibration of the overall model under a broad class of distribution shifts in hard-routed models, but is insufficient for calibrating soft-routed models. To address this, we propose an adversarial reweighting that penalizes calibration errors of the routed aggregate under distribution shift, and we demonstrate that it improves the accuracy-calibration tradeoff both on average and on difficult subsets of the data, across model classes, prediction tasks, and distribution shifts.

## 1. Introduction

Mixture-of-experts (MoE) is a widely used architecture that decomposes a difficult learning problem into simpler, specialized subproblems. In an MoE, a routing model directs inputs to a single expert under hard routing, or to a weighted combination of experts under soft routing. This design naturally supports *conditional computation*, where only a subset of parameters are activated per input. Conditional computation allows MoEs to scale model capacity without a proportionate increase in compute (Fedus et al., 2022; Du et al., 2022; Ludziejewski et al., 2025), while also learning

specialized experts for heterogeneous data (Jacobs et al., 1991; Dai et al., 2024; Guo et al., 2026), leading to renewed interest in MoEs for large-scale systems (Dai et al., 2024; Lepikhin et al., 2021; Jiang et al., 2024).

Because MoEs decompose prediction across experts that share the input space, there is a common intuition that they should be naturally robust to shifts in how that space is populated. For example, if an MoE contains a reliable math expert and a reliable history expert, one might expect the overall predictor to remain reliable, even if the training data emphasizes history questions while the test data emphasizes math questions. In this work, *we show that this intuition is misleading*: even when every expert is individually reliable, the aggregate predictor of a soft-routed MoE can become systematically unreliable under distribution shift.

More precisely, we show that when routing patterns differ between training and test, a soft-routed MoE can become miscalibrated, even if every expert remains perfectly calibrated on its own routing-induced view of the data. This failure therefore cannot be explained by miscalibrated experts, by standard covariate shift, or by incorrect conditional likelihoods. Instead, it arises from a mismatch in how often different routing configurations appear at training versus test time: configurations that balanced one another at a given confidence value under training can fall out of balance under the test distribution, even though each expert keeps making the same predictions on the same inputs.

To understand this phenomenon, we analyze how calibration interacts with routing under distribution shift. In hard-routed MoEs, expert-level calibration is sufficient to guarantee robustness to changes in how frequently different inputs appear: because routing partitions the input space, shifts in region prevalence do not affect calibration. In contrast, soft-routed MoEs replace explicit partitions with overlapping, routing-weighted views of the data. The aggregate prediction is then determined by the full joint configuration of routing weights and expert outputs, and aggregate calibration holds only because the configurations that share a confidence value happen to balance each other under the training distribution. When the test distribution shifts how often each configuration appears, that balance is lost and aggregate calibration breaks, even though every expert remains exactly as reliable as before.

---
[1]Johns Hopkins University, Baltimore, MD, USA. Correspondence to: Gina Wong <gwong15@jh.edu>.

*Proceedings of the 43$^{rd}$ International Conference on Machine Learning*, Seoul, South Korea. PMLR 306, 2026. Copyright 2026 by the author(s).

We propose to address this fragility by training the aggregate predictor against adversarial reweightings of the training distribution. The central object in our method is an *entropy-balanced adversarial reweighting*: among reweighted versions of the training distribution that put pressure on the current predictor, we choose the one closest to the empirical distribution in relative entropy. This entropy-balancing principle yields a tilted softmax over per-example proper losses, so that high-loss examples—the examples most likely to be confidently wrong or miscalibrated—receive more weight, while the reweighting remains smooth and diffuse rather than collapsing to a hard subset. The resulting objective focuses training on the routing-overlap regions where experts disagree and the router is unstable, without requiring explicit knowledge of the test-time distribution shift.

To summarize our main contributions:

- We characterize the distribution shifts under which calibration is preserved in hard- and soft-routed MoEs.

- We show that soft-routed MoEs are not robust to shifts in routing-induced expert weighting, even when all the experts are individually calibrated.

- We propose an entropy-balanced adversarial reweighting of a proper scoring loss that directly regularizes the aggregate MoE predictor on high-loss routing-overlap regions, and we relate it to entropic DRO, CVaR-style tail risk, and robust multiaccuracy as alternative views of the same adversarial-reweighting principle.

- We show that this method improves calibration under artificial and natural distribution shifts across image, domain-generalization, and text benchmarks, often with little or no accuracy cost.

## 2. Background

**Calibration and proper scoring rules**  Calibration formalizes the idea that a model's predictive probabilities should match empirical frequencies (Dawid, 1982). For a probabilistic binary classifier $f : \mathcal{X} \to [0,1]$, marginal calibration requires

$$\mathbb{P}(Y = 1 \mid f(X) = p) = p \quad \text{for all } p \in [0,1].$$

That is, among examples where the model assigns probability $p$, the positive-label frequency should be $p$.

A standard way to train probabilistic predictors is to optimize a strictly proper scoring rule. A scoring rule $\ell(f(x), y)$ is *strictly proper* if its expected value is uniquely minimized when $f(x)$ equals the true conditional probability $\mathbb{P}(Y = 1 \mid X = x)$ (Gneiting and Raftery, 2007). Common examples include cross-entropy (log loss) and the Brier score. We write $L_\theta(X, Y) = \ell(f_\theta(X), Y)$ for a proper loss.

**Mixture-of-experts**  A mixture-of-experts (MoE) model decomposes prediction into a set of experts combined with a routing mechanism (Jacobs et al., 1991). Let $\{f_k\}_{k=1}^K$ be experts with outputs $f_k(x) \in [0,1]$, and let $r(x) = (r_1(x), \ldots, r_K(x)) \in \Delta^{K-1}$ be routing weights on the $(K-1)$-dimensional probability simplex. The aggregate prediction is

$$f(x) = \sum_{k=1}^K r_k(x) f_k(x).$$

Routing may be hard or soft. Under hard routing, $r(x)$ is one-hot, so each input is assigned to a single expert and the input space is partitioned into routing regions. Under soft routing, multiple experts may receive positive weight on the same input, so experts are trained and evaluated on overlapping, routing-weighted views of the data.

We study MoEs whose experts are calibrated on the data views they receive. This assumption lets expert predictions be interpreted as conditional expectations and isolates the effect of routing: the failures we analyze are not caused by individually unreliable experts, but by how calibrated experts are combined. Proper losses such as cross-entropy loss or the Brier score naturally encourage this form of expert-level calibration under the training distribution. Related empirical evidence appears in mixture-of-calibrated-experts models, where calibrating experts before combining them improves aggregate performance (Oksuz et al., 2024; Roschewitz et al., 2025).

Our theoretical analysis is stated for binary prediction, while the multiclass experiments evaluate the standard scalar notion of *confidence calibration*. With expert predictions $p_k(x) \in \Delta^{C-1}$ and aggregate prediction $p(x) = \sum_k r_k(x) p_k(x)$, let $\hat{y}(x) = \arg\max_y p_y(x)$ and $c(x) = \max_y p_y(x)$; confidence calibration requires $\mathbb{P}(Y = \hat{y}(X) \mid c(X) = q) = q$ for all $q$. The configuration-collapse phenomenon analyzed in §4 carries over through this scalar confidence: distinct routing/expert configurations can induce the same $c(x)$ while having different correctness frequencies, so reweighting configurations within a confidence level set can change aggregate confidence calibration.

**Distribution shift through routing**  Distribution shift refers to the common phenomenon where the training and test distributions differ. In this paper, we focus on shifts where MoE reweightings affect the routing mechanism; in hard-routed MoEs, these correspond to changes in the prevalence of routing regions, and in soft-routed MoEs, they correspond to changes in the prevalence of routing-weighted overlap configurations. As a result, shifts that merely change how often different regions appear can be harmless under hard routing but harmful under soft routing, because the aggregate predictor depends on the stability of the routing weights themselves.

# 3. The simple robustness of hard routing

We begin with MoEs under *hard routing*. This setting provides a useful baseline because the router assigns each input to a single expert, partitioning the input space. As a result, calibration of the aggregate predictor is governed by a simple statistic: the identity of the selected expert and the confidence reported by that expert. This statistic forms an information bottleneck for calibration, which makes hard-routed MoEs robust to a natural class of distribution shifts.

## 3.1. The expert-confidence statistic

In the hard routing setting, the routing vector $r(x)$ is one-hot. We write $h(x) = \arg\max_k r_k(x)$ for the deterministic expert assignment, and define the corresponding routing regions $R_k := \{x \in \mathcal{X} : h(x) = k\}$ for $k = 1, \ldots, K$. The hard-routed MoE predictor is therefore piecewise, that is, $f(x) = f_{h(x)}(x)$.

For calibration, the relevant object is not the full input $X$, but the selected expert together with its reported confidence. Define
$$S_{\text{hard}}(X) := \big(h(X),\, f_{h(X)}(X)\big). \tag{1}$$

Under hard routing, the aggregate prediction is fully determined by this statistic. Expert-level calibration can be written as
$$\mathbb{P}_{\text{train}}(Y = 1 \mid X \in R_k,\, f_k(X) = p) = p$$

for all $k$ and $p \in [0, 1]$, or equivalently,
$$\mathbb{P}_{\text{train}}(Y = 1 \mid S_{\text{hard}}(X) = (k, p)) = p$$

for all $k$ and $p \in [0, 1]$. Thus, under hard routing, calibration depends only on the expert-confidence pair $(k, p)$. Once we condition on $S_{\text{hard}}(X)$, any other features of $X$ are irrelevant to calibration. This is the sense in which hard routing induces an information bottleneck.

## 3.2. Calibration-preserving shifts under hard routing

The bottleneck view gives a direct characterization of distribution shifts that preserve calibration. Suppose the hard-routed MoE is calibrated under the training distribution. Then it remains calibrated under any test distribution satisfying
$$P_{\text{test}}(Y \mid S_{\text{hard}}(X)) = P_{\text{train}}(Y \mid S_{\text{hard}}(X)). \tag{2}$$

Indeed, if $S_{\text{hard}}(X) = (k, p)$, then (2) implies
$$\mathbb{P}_{\text{test}}(Y = 1 \mid S_{\text{hard}}(X) = (k, p))$$
$$= \mathbb{P}_{\text{train}}(Y = 1 \mid S_{\text{hard}}(X) = (k, p)) = p.$$

Since $f(X) = p$ whenever $S_{\text{hard}}(X) = (k, p)$, the aggregate hard-routed predictor remains calibrated under the test distribution.

A simple and important special case is reweighting of routing regions. If the test distribution is a mixture of the region-conditional training distributions,
$$P_{\text{test}}(\cdot) = \sum_{k=1}^{K} \alpha_k\, P_{\text{train}}(\cdot \mid X \in R_k),$$

where $\alpha \in \Delta^{K-1}$ gives the new prevalence of the routing regions, then the conditional distribution within each expert-confidence slice is unchanged. Consequently, changing how often different experts are selected does not by itself affect calibration.

More generally, hard routing is robust to any shift that preserves the label distribution within each expert-confidence slice. Such shifts may change the marginal frequency of experts, the marginal frequency of confidence levels, or the covariate distribution inside a routing region, as long as they do not change $P(Y \mid S_{\text{hard}}(X))$. This is the specific robustness provided by hard routing: calibration is insensitive to changes outside the expert-confidence bottleneck.

# 4. The fragility of soft routing

We now turn to *soft routing*, where the router assigns each input a distribution over experts rather than a single expert. Soft routing is more expressive, but removes the discrete expert-confidence bottleneck that made the hard routing case more simple. Experts now receive overlapping, routing-weighted views of the same examples, so calibrating each expert on its own view no longer calibrates the aggregate predictor.

## 4.1. Expert calibration gives only a marginal constraint

Under soft routing, each expert may receive nonzero weight on the same input. Expert $k$ therefore does not see a disjoint region $R_k$; it sees the training distribution reweighted by its routing weight $r_k(X)$. This motivates a weighted notion of expert calibration. We say that expert $k$ is calibrated on its routing-weighted view if
$$\mathbb{E}_{P_{\text{train}}}[r_k(X)\,(Y - f_k(X)) \mid f_k(X) = p] = 0$$

for all $p \in [0, 1]$. Intuitively, this condition says that expert $k$ is reliable on the examples it receives, counted in proportion to the weight assigned by the router; equivalently, expert $k$ is calibrated under the routing-weighted distribution $Q_k$ with $dQ_k \propto r_k(X)\, dP_{\text{train}}$. The important limitation of this condition is that it is indexed by each expert's own confidence, not by the final mixture confidence, so it says nothing yet about how the experts' residuals align after the router averages them.

Integrating this condition over $p$ and summing over $k$ gives

$$\sum_{k=1}^{K} \mathbb{E}_{P_{\text{train}}}[r_k(X)(Y - f_k(X))] = \mathbb{E}_{P_{\text{train}}}[Y - f(X)] = 0.$$

So expert calibration forces only a *marginal* mean balance on the aggregate predictor; it does not force the level-set condition $\mathbb{E}[Y - f(X) \mid f(X) = p] = 0$ that aggregate calibration requires. This gap is the source of the fragility we now describe.

## 4.2. Aggregation collapses distinct routing configurations

Under soft routing, the aggregate prediction is produced by the full joint configuration of routing weights and expert outputs,

$$S_{\text{soft}}(X) := \left( \{r_k(X)\}_{k=1}^{K}, \{f_k(X)\}_{k=1}^{K} \right), \qquad (3)$$

but the scalar prediction $f(X) = \sum_k r_k(X) f_k(X)$ is a many-to-one function of this configuration—that is, two inputs can share the same aggregate prediction while arising from very different configurations.

This distinction is what matters for calibration. Let

$$m(s) := \mathbb{E}_{P_{\text{train}}}[Y \mid S_{\text{soft}}(X) = s]$$

be the outcome frequency of a configuration. Aggregate calibration at level $p$ requires

$$\mathbb{E}_{P_{\text{train}}}[Y \mid f(X) = p] = p.$$

But since the event $\{f(X) = p\}$ may contain many different configurations $s$, this condition averages over all configurations that yield the same scalar prediction:

$$\mathbb{E}_{P_{\text{train}}}[Y \mid f(X) = p]$$
$$= \mathbb{E}_{P_{\text{train}}}[m(S_{\text{soft}}(X)) \mid f(X) = p].$$

Aggregate calibration can therefore hold because configuration-level deviations cancel within a level set, not because they vanish configuration by configuration.

For example, suppose two configurations $s$ and $s'$ both yield the same aggregate prediction $p$, but have different conditional outcome frequencies:

$$m(s) = p + \delta, \qquad m(s') = p - \delta.$$

If they appear in balancing proportions under the training distribution, the aggregate predictor is calibrated at level $p$. But a test-time shift that changes the relative prevalence of $s$ and $s'$ changes the label frequency among examples predicting $p$, and breaks calibration—even though each expert remains calibrated on its own routing-weighted view.

Expert calibration is therefore not enough: it constrains each expert on its own weighted stream, but it does not force all configurations that collapse to the same aggregate prediction to have the same outcome frequency. The following proposition makes this precise for shifts that change the prevalence of configurations while preserving the label distribution given a configuration. Write $S = S_{\text{soft}}(X)$, and recall that $f(X) = \sum_k r_k(X) f_k(X)$ is a deterministic function of $S$ and that $m(s) = \mathbb{E}_{P_{\text{train}}}[Y \mid S = s]$.

**Proposition 4.1** (Aggregate calibration under configuration reweighting)**.** *Consider a test distribution that reweights routing configurations by a density $a(S) \geq 0$ with $\mathbb{E}_{P_{train}}[a(S)] = 1$, while preserving the conditional $Y \mid S$. Then, with conditional expectations understood in the regular-conditional sense, for almost every prediction level $p$ under the training pushforward law of $f(X)$ with $\mathbb{E}_{P_{train}}[a(S) \mid f(X) = p] > 0$,*

$$\mathbb{E}_{P_{test}}[Y - f(X) \mid f(X) = p]$$
$$= \frac{\mathbb{E}_{P_{train}}[a(S)(m(S) - p) \mid f(X) = p]}{\mathbb{E}_{P_{train}}[a(S) \mid f(X) = p]}. \qquad (4)$$

*Aggregate calibration is preserved under every such reweighting if and only if $m(S) = f(X)$ almost surely; equivalently, for almost every prediction level $p$, $m(S) = p$ almost surely under the regular conditional law of $S$ given $f(X) = p$.*

Ordinary training-distribution calibration is the $a \equiv 1$ case of (4), which asks only that the deviations $m(S) - p$ cancel on average within each level set. The proposition shows that calibration survives *arbitrary* configuration reweighting exactly when these deviations vanish configuration by configuration.

The failure is most pronounced in routing-overlap regions where experts receive non-negligible weight and disagree, so that the aggregate prediction depends sensitively on the routing weights. We call shifts that alter the prevalence of such configurations *routing-induced reweighting*: they are the soft-routing analogue of hard-routing region reweighting, but with a critical difference that they can break aggregate calibration without changing the reliability of any expert. The condition that rules them out, $m(S) = f(X)$, is far stronger than its hard-routing counterpart (2): it conditions on the entire vector of routing weights and expert predictions, not merely on an expert identity and a confidence value, so the class of shifts certifiably preserving calibration is correspondingly narrower and less interpretable.

## 5. Robust training against routing-induced reweighting

Proposition 4.1 identifies the problematic shifts as reweightings of soft-routing configurations, with calibration fail-

ure occurring where $m(S) \neq f(X)$. As the test-time density $a(S_{\text{soft}}(X))$ is unobserved, and $S_{\text{soft}}(X)$ is a high-dimensional, model-induced statistic rather than a labeled group variable, we cannot target this condition directly. We therefore need an observable feature of an example that correlates with $m(S) \neq f(X)$. The per-example proper loss is the natural candidate, as it is large precisely on examples where the aggregate predictor is poorly aligned with the observed label.

Let $L_\theta(X, Y) := \ell(f_\theta(X), Y)$, where $\ell$ is a strictly proper scoring loss and $\theta$ contains the router and expert parameters. A shift absolutely continuous with respect to $P_{\text{train}}$ can be represented by a density ratio $w(X, Y) \geq 0$ with $\mathbb{E}_{P_{\text{train}}}[w] = 1$. This motivates the robust scoring objective

$$\min_\theta \sup_{w \in \mathcal{W}} \mathbb{E}_{P_{\text{train}}}[w(X, Y) L_\theta(X, Y)], \qquad (5)$$

where $\mathcal{W}$ specifies which reweightings the adversary may choose.

In our setting, the concern is not that one fixed group will dominate at test time, but that routing-induced reweighting will change the relative mass of broad overlap regions. We therefore want the adversary to stress high-loss examples while changing the empirical distribution as little as possible. Entropy balancing gives this least-distorted adversary. On a minibatch with losses $L_i = L_\theta(x_i, y_i)$ and uniform weights $u_i = 1/n$, it solves

$$\min_{q \in \Delta_n} \text{KL}(q \| u) \quad \text{subject to} \quad \sum_i q_i L_i = c,$$

for an adversarial loss level $c$ above the empirical mean. The solution, derived in Appendix B.1, is the exponential tilt

$$q_i^\eta = \frac{\exp(\eta L_i)}{\sum_{j=1}^n \exp(\eta L_j)},$$

where $\eta \geq 0$ controls the strength of the reweighting. Thus high-loss examples receive more weight, but every example keeps positive mass.

We propose two objectives based on this adversary. *Robust MoE* applies entropy-balanced reweighting to the full minibatch, yielding a smooth robust analogue of ERM. *Robust Filtered* applies the same reweighting only to a routing-relevant subset, while retaining an ERM term on the full minibatch so that all examples continue to update the model.

**Robust MoE**   Robust MoE trains on the expected loss under this entropy-balanced adversary:

$$\mathcal{L}_{\text{R-MoE}}(\theta) = \sum_{i=1}^n q_i^\eta L_i. \qquad (6)$$

A useful identity is

$$\mathcal{L}_{\text{R-MoE}}(\theta) = \rho_\eta(L) + \frac{1}{\eta} \text{KL}(q^\eta \| u), \qquad (7)$$

where $\rho_\eta(L) = \frac{1}{\eta} \log \frac{1}{n} \sum_i e^{\eta L_i}$. The first term is the standard entropic risk; the second is an induced concentration penalty that grows when the current losses concentrate the adversary on a small set, which is exactly the regime a small collection of routing-overlap configurations can dominate the robust objective.

This decomposition also shows that Robust MoE is a conservative surrogate for distributionally robust optimization over a KL ball. For the uncertainty set $\{q : \text{KL}(q\|u) \leq \varepsilon\}$ on the minibatch, the standard dual bound gives, for any $\eta > 0$,

$$\sup_{q:\text{KL}(q\|u)\leq\varepsilon} \sum_i q_i L_i \leq \rho_\eta(L) + \frac{\varepsilon}{\eta} \leq \mathcal{L}_{\text{R-MoE}}(\theta) + \frac{\varepsilon}{\eta}.$$

Minimizing the training objective therefore minimizes an upper bound on the worst-case reweighted loss within the ball; the pair $(\eta, \varepsilon)$ controls how aggressively this certificate allows the adversary to concentrate. In the limit $\eta \downarrow 0$, $q^\eta$ becomes uniform and the training objective reduces to ERM, while larger $\eta$ moves the objective toward higher-loss examples.

The same bound holds at the population level, where it controls the actual test risk under an absolutely continuous shift. Let $\mathcal{L}_{\text{R-MoE}}^{\text{pop}}(\theta) := \mathbb{E}_{q^\eta}[L_\theta]$ denote the population tilted mean, with $dq^\eta \propto e^{\eta L_\theta} dP_{\text{train}}$. Then, for any shifted distribution $dP_w = w \, dP_{\text{train}}$,

$$\begin{aligned} \mathbb{E}_{P_w}[L_\theta] &\leq \tfrac{1}{\eta} \log \mathbb{E}_{P_{\text{train}}}\big[e^{\eta L_\theta}\big] + \tfrac{1}{\eta}\text{KL}(P_w\|P_{\text{train}}) \\ &\leq \mathcal{L}_{\text{R-MoE}}^{\text{pop}}(\theta) + \tfrac{1}{\eta}\text{KL}(P_w\|P_{\text{train}}). \end{aligned} \qquad (8)$$

This is a value-level certificate: it controls robust-risk values over a reweighting family and is the population object used in the multiaccuracy bound of Appendix C. It does not claim that each gradient step is itself a multiaccuracy correction.

**Robust Filtered**   Not every high-loss example reflects routing fragility; some examples are intrinsically ambiguous, mislabeled, or difficult for all experts. Robust Filtered therefore applies entropy-balanced reweighting only to a routing-relevant subset $A$, while keeping an ERM term over the full minibatch. In the experiments, $A$ is the union of examples where the mixture incurs higher loss than the best expert and examples where the experts disagree substantially around the mixture prediction; the exact criteria are given in Appendix D.3. The objective is

$$\mathcal{L}_{\text{RF-MoE}}(\theta) = \frac{1}{n} \sum_{i=1}^n L_i + \sum_{i \in A} q_{i,A}^\eta L_i,$$

where

$$q_{i,A}^\eta = \frac{\exp(\eta L_i)}{\sum_{j \in A} \exp(\eta L_j)}.$$

The ERM term keeps all examples active, while the robust term puts additional pressure on the routing-sensitive examples most directly tied to the failure mode in §4.

Appendix B derives the entropy-balanced weights and compares them with entropic DRO and CVaR-style tail risk. Appendix C gives the corresponding multiaccuracy view: the worst-case squared residual over the reweighting family upper-bounds residual alignment with routing-weighted auditors of the form $r_k(x)\phi(f(x))$.

## 6. Experiments

### 6.1. Datasets and models

All models share the same MoE structure. A shared backbone (ResNet-18 or DistilBERT) produces a feature representation, which is passed to an MoE classification head consisting of $K=4$ expert linear classifiers and a 2-layer MLP router with softmax outputs. The entire model—backbone, experts, and router—is trained end-to-end.

We evaluate on three dataset–backbone pairs spanning image classification, domain generalization, and text toxicity detection; convolutional and transformer backbones; and both artificial and natural distribution shifts.

- **CIFAR-10H** (Peterson et al., 2019) extends CIFAR-10 with human agreement annotations. We pair it with a ResNet-18 trained from scratch. Images where annotators disagree simulate routing ambiguity: we define a "hard" subset as containing images with low human agreement, creating an artificial shift that stresses routing-overlap regions.
- **PACS** (Li et al., 2017) contains four visually distinct domains (Photo, Art Painting, Cartoon, Sketch). We use ResNet-18 pretrained on ImageNet and follow leave-one-domain-out evaluation, training on three domains and testing on the held-out target. Each held-out domain constitutes a natural domain/style shift.
- **CivilComments** (Koh et al., 2021) is a text toxicity classification benchmark from WILDS. We pair this with a DistilBERT backbone pretrained on English text. Comments mentioning demographic identity groups form a natural subpopulation shift, as toxicity classifiers tend to have high false positive rates on benign mentions of demographic groups.

Full dataset descriptions and model architecture details are given in Appendix D.2.

### 6.2. Baselines

We compare **Robust MoE** and **Robust Filtered** against the following baselines:

- **Single Expert**: a standard classifier (same backbone, no routing) trained with cross-entropy.
- **Vanilla MoE**: a soft-routed MoE trained with ERM on cross-entropy.
- **MoCaE** (Oksuz et al., 2024; Roschewitz et al., 2025): an MoE where experts are individually calibrated after training, before their predictions are combined. This directly tests whether calibrating experts individually is sufficient to fix mixture-level miscalibration.
- **Frequency-aware Gradient Rectification (FGR)** (Zhang et al., 2025): FGR is a training procedure designed to improve calibration under covariate shift. It was designed for single models under image corruptions, not for MoE routing fragility, but it provides a strong calibration-aware baseline.

We also report **FGR + Robust** as a composition of FGR with our robust objective. FGR operates on a different axis than Robust MoE. Robust MoE modifies the *training objective*, while FGR modifies the *gradient update rule*; these are independent design choices that can be freely combined. We create FGR + Robust by using the Robust MoE loss as FGR's main objective: the classification gradient is computed from the full entropy-balanced objective on the frequency-perturbed batch, while the calibration gradient (soft ECE on the original batch) and the rectification logic are unchanged.

We additionally report results with post-training temperature scaling (TS) to separate improvements in calibration due to training from those achievable by post-training calibration alone. More details on how we implement all the baselines are given in Appendix D.1.

### 6.3. Results

The experiments are designed to test three empirical claims suggested by our analysis: (i) soft-routed MoEs can remain miscalibrated even when expert-level calibration is improved; (ii) adversarial reweighting of the aggregate proper loss improves calibration under routing-induced shifts; and (iii) focusing the robust term on routing-relevant examples improves the accuracy–calibration tradeoff.

Results on CIFAR-10H are reported in Table 1, where Accuracy, ECE, and ECE+TS (ECE after aggregate temperature scaling) are reported on all images and on the hard subset of low-agreement images. Results on CivilComments in Table 3 follow the same convention, except that the hard subset consists of comments mentioning demographic identities. Table 2 reports PACS results for each leave-one-domain-out target domain. Additional dataset details and hard-subset definitions are given in §6.1 and Appendix D.2. All standard errors are computed over five random seeds.

We provide CIFAR-10H reliability diagrams in Figure 1.

*Table 1.* CIFAR-10H: accuracy and ECE on all images and on the hard subset of images with low human agreement

| Method | Accuracy | Hard Accuracy | ECE | ECE+TS | Hard ECE | Hard ECE+TS |
|---|---|---|---|---|---|---|
| Single Expert | **0.922 ± 0.001** | **0.648 ± 0.014** | 0.054 ± 0.001 | 0.048 ± 0.001 | 0.276 ± 0.018 | 0.256 ± 0.017 |
| Vanilla MoE | 0.920 ± 0.001 | 0.637 ± 0.019 | 0.055 ± 0.001 | 0.049 ± 0.001 | 0.281 ± 0.019 | 0.262 ± 0.020 |
| MoCaE | 0.920 ± 0.001 | 0.637 ± 0.019 | 0.049 ± 0.001 | 0.042 ± 0.002 | 0.262 ± 0.020 | 0.241 ± 0.019 |
| FGR | 0.919 ± 0.002 | 0.643 ± 0.010 | 0.051 ± 0.001 | 0.045 ± 0.001 | 0.262 ± 0.015 | 0.243 ± 0.016 |
| Robust MoE | 0.904 ± 0.001 | 0.612 ± 0.012 | 0.090 ± 0.019 | 0.036 ± 0.010 | 0.074 ± 0.018 | 0.115 ± 0.017 |
| Robust Filtered | 0.910 ± 0.003 | 0.624 ± 0.015 | **0.013 ± 0.004** | **0.007 ± 0.001** | 0.122 ± 0.017 | 0.139 ± 0.017 |
| FGR + Robust | 0.903 ± 0.003 | 0.600 ± 0.015 | 0.083 ± 0.010 | 0.028 ± 0.003 | **0.065 ± 0.012** | **0.108 ± 0.013** |

*Table 2.* PACS: accuracy and ECE on each leave-one-out target domain

| Method | Photo | | | Art | | | Cartoon | | | Sketch | | |
|---|---|---|---|---|---|---|---|---|---|---|---|---|
| | Acc | ECE | ECE+TS | Acc | ECE | ECE+TS | Acc | ECE | ECE+TS | Acc | ECE | ECE+TS |
| Single Expert | **.943**±.004 | .031±.003 | .024±.003 | **.780**±.006 | .128±.005 | .106±.006 | .730±.013 | .166±.012 | .140±.012 | .637±.025 | .217±.030 | .230±.030 |
| Vanilla MoE | .941±.002 | .033±.002 | .026±.001 | .767±.007 | .141±.006 | .118±.007 | **.732**±.013 | .171±.011 | .146±.010 | .667±.009 | .183±.017 | .189±.011 |
| MoCaE | .941±.002 | .027±.002 | .018±.001 | .767±.007 | .119±.007 | .094±.007 | **.732**±.013 | .147±.010 | .119±.009 | .667±.009 | .190±.012 | .195±.009 |
| FGR | .940±.002 | .035±.002 | .027±.003 | .768±.008 | .143±.007 | .122±.007 | .728±.010 | .175±.013 | .150±.013 | .667±.012 | .176±.020 | .187±.018 |
| Robust MoE | .931±.006 | .018±.005 | .019±.005 | .692±.023 | .088±.025 | .071±.023 | .723±.010 | **.108**±.018 | **.088**±.021 | .666±.033 | **.033**±.017 | **.072**±.025 |
| Robust Filtered | .930±.006 | **.016**±.005 | **.013**±.004 | .732±.018 | .075±.013 | .057±.015 | .718±.018 | .135±.022 | .113±.023 | **.670**±.020 | .065±.030 | .093±.028 |
| FGR + Robust | .928±.005 | **.016**±.005 | .017±.006 | .732±.017 | **.061**±.027 | **.045**±.023 | .698±.024 | .134±.030 | .113±.032 | .660±.023 | .072±.025 | .105±.026 |

*Table 3.* CivilComments: accuracy and ECE on all comments and on the hard subset of comments with demographic identities

| Method | Accuracy | Hard Accuracy | ECE | ECE+TS | Hard ECE | Hard ECE+TS |
|---|---|---|---|---|---|---|
| Single Expert | 0.913 ± 0.002 | 0.867 ± 0.002 | 0.069 ± 0.001 | 0.064 ± 0.001 | 0.105 ± 0.001 | 0.097 ± 0.001 |
| Vanilla MoE | 0.912 ± 0.001 | 0.866 ± 0.002 | 0.071 ± 0.001 | 0.065 ± 0.001 | 0.108 ± 0.002 | 0.100 ± 0.002 |
| MoCaE | 0.912 ± 0.001 | 0.866 ± 0.002 | 0.066 ± 0.002 | 0.059 ± 0.002 | 0.101 ± 0.002 | 0.091 ± 0.002 |
| FGR | 0.906 ± 0.003 | 0.857 ± 0.005 | 0.044 ± 0.004 | 0.035 ± 0.004 | 0.065 ± 0.006 | 0.053 ± 0.006 |
| Robust MoE | 0.914 ± 0.000 | 0.868 ± 0.000 | **0.025 ± 0.001** | **0.018 ± 0.001** | **0.037 ± 0.002** | **0.029 ± 0.002** |
| Robust Filtered | **0.915 ± 0.002** | **0.869 ± 0.004** | 0.027 ± 0.004 | 0.021 ± 0.003 | 0.040 ± 0.005 | 0.031 ± 0.004 |
| FGR + Robust | 0.903 ± 0.004 | 0.853 ± 0.006 | 0.054 ± 0.008 | 0.047 ± 0.003 | 0.065 ± 0.010 | 0.057 ± 0.005 |

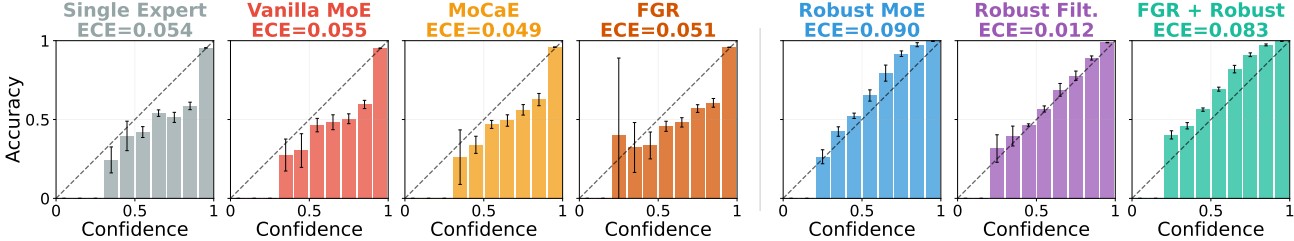

*Figure 1.* Reliability diagrams for all methods on CIFAR-10H. Each panel compares predicted confidence with empirical accuracy; perfect confidence calibration lies on the diagonal. Non-robust baselines concentrate much of their mass in high-confidence bins, while the robust objectives spread predictions over a wider confidence range and better align confidence with accuracy. Additional reliability diagrams for the remaining datasets and hard subsets appear in Appendix E.

Additional reliability diagrams for the remaining datasets and hard subsets appear in Appendix E.

**Takeaway 1: robust training improves calibration where routing is stressed.** The clearest gains appear on the shifted or ambiguous subsets that most directly stress the soft router. On CIFAR-10H, the Vanilla MoE and MoCaE have hard-subset ECEs of 0.281 and 0.262, comparable to the Single Expert value of 0.276. Robust MoE reduces this to 0.074, and FGR+Robust further reduces it to 0.065. On CivilComments, Robust MoE and Robust Filtered reduce hard-subset ECE from 0.108 for Vanilla MoE and 0.101 for MoCaE to 0.037 and 0.040, while slightly improving hard-subset accuracy. On PACS, a robust method or robust composition obtains the lowest ECE in every held-out target domain: Robust Filtered is strongest on Photo, FGR+Robust

is strongest on Art, and Robust MoE is strongest on Cartoon and Sketch.

**Takeaway 2: calibrating experts individually is not enough.** MoCaE is an important diagnostic baseline because it post-hoc calibrates each expert with its own temperature before they are mixed. Because MoCaE only rescales each expert's probabilities and does not change the underlying mixture model, its accuracy is identical to Vanilla MoE in every cell of every table. Per-expert calibration does deliver a modest ECE improvement over Vanilla on the full distribution (e.g., CIFAR-10H ECE goes from 0.055 to 0.049, CivilComments ECE from 0.071 to 0.066), and a similarly modest improvement on hard subsets (CIFAR-10H Hard ECE goes from 0.281 to 0.262, CivilComments Hard ECE from 0.108 to 0.101). These improvements are

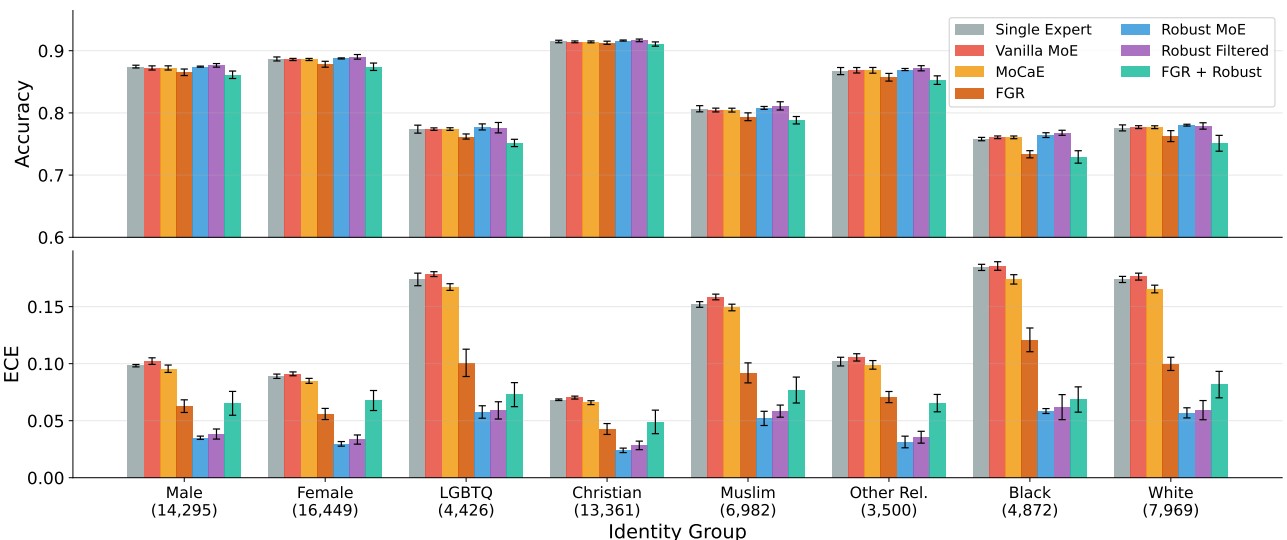

*Figure 2.* Accuracy and ECE on the comments mentioning demographic identity groups in CivilComments, which are prone to higher false positive rates in toxicity classification. The top panel reports accuracy and the bottom panel reports ECE, with subgroup sample sizes shown below each label. Robust methods reduce ECE while maintaining competitive accuracy, mirroring the aggregate gains in Table 3.

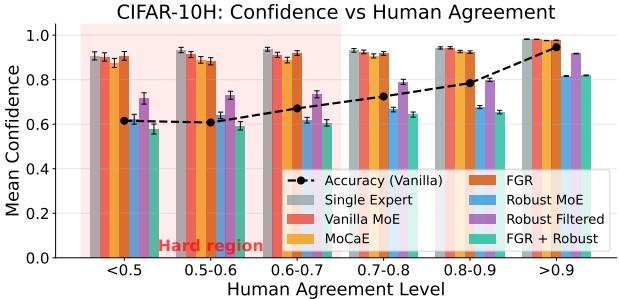

*Figure 3.* Accuracy and mean confidence on the CIFAR-10H hard subset by human agreement level. Lower agreement indicates more ambiguous images; the dashed line shows Vanilla MoE accuracy. Robust methods lower confidence as ambiguity increases.

small relative to improvement from the robust methods—on CIFAR-10H and CivilComments, Robust MoE achieves Hard ECEs of 0.074 and 0.037. Per-expert temperature scaling helps improves per-expert overconfidence but does not close the mixture-level miscalibration that arises from routing-induced reweighting §4.

**Takeaway 3: Robust Filtered gives a better accuracy–calibration tradeoff than full reweighting in several settings.** The full Robust MoE objective puts additional weight on all high-loss examples, which can improve calibration sharply but may overemphasize examples that are hard for reasons unrelated to routing. This is visible on CIFAR-10H and PACS-Art, where Robust MoE lowers calibration error but loses accuracy relative to the strongest non-robust classifier. Robust Filtered mitigates this tradeoff by applying the robust term only to routing-relevant examples while retaining an ERM term on the full minibatch.

It achieves the best overall CIFAR-10H ECE (0.013), the best CivilComments accuracy (0.915), and the best PACS-Sketch accuracy (0.670), while maintaining much lower ECE than the non-robust MoE baselines.

**Takeaway 4: post-hoc temperature scaling helps, but does not explain the gains.** Temperature scaling reduces global ECE for several methods, but the robust objectives remain strongest after temperature scaling on the hard or shifted evaluations. On CIFAR-10H hard examples, the best non-robust ECE+TS is 0.241 from MoCaE, whereas Robust MoE and FGR+Robust achieve 0.115 and 0.108. On CivilComments hard examples, Robust MoE and Robust Filtered achieve ECE+TS values of 0.029 and 0.031, compared with 0.100 for Vanilla MoE, 0.091 for MoCaE, and 0.053 for FGR. Thus the improvement is not merely a post-hoc rescaling of confidence; the training objective changes which examples receive probability mass and how the model behaves on shifted subpopulations.

The reliability diagrams in Figure 1 show the same qualitative pattern. The non-robust baselines are highly concentrated in high-confidence bins, and the sparsely populated low-confidence bins have larger error bars. In contrast, the robust methods and robust composition spread predictions over a wider confidence range and align more closely with the diagonal, indicating that uncertainty is being expressed in the regions where the model is less reliable.

**Subgroup behavior is consistent with the routing-fragility mechanism.** We bin CIFAR-10H images by human agreement level in Figure 3. The non-robust baselines maintain mean confidence near 0.9 even as human agreement decreases, including for the lowest-agreement images

where Vanilla MoE accuracy is roughly 0.6. In contrast, Robust MoE, Robust Filtered, and FGR+Robust reduce confidence as human agreement decreases.

On CivilComments, we evaluate calibration and accuracy across the eight demographic identity groups present in the comments: Male, Female, LGBTQ, Christian, Muslim, Other Religions, Black, and White (Figure 2). Comments can belong to multiple groups. Across these groups, the robust methods consistently reduce ECE relative to the non-robust MoE baselines while maintaining comparable accuracy. This indicates that the aggregate improvements in Table 3 are not driven by a single identity group, but by a broader correction across shifted subpopulations.

## 7. Related work

**MoEs, calibration, and distribution shift.** Most work on MoEs under distribution shift use specialization to *mitigate* shift: domain-specific or skill-diverse experts support target-domain adaptation, long-tailed recognition, and open-set domain adaptation, where routing patterns can also signal unfamiliar inputs (Zhong et al., 2022; Wang et al., 2023; Du et al., 2025). A separate line calibrates experts before combining them (Oksuz et al., 2024; Roschewitz et al., 2025). Our view differs on both counts: the aggregation can become miscalibrated under shift even when every expert is calibrated, and our MoCaE baseline shows that per-expert calibration does not remove this aggregate failure (§6).

**Calibration, multicalibration, and multiaccuracy.** Calibration is a classical requirement for probabilistic prediction (Dawid, 1982; Gneiting and Raftery, 2007), and modern neural predictors are often evaluated or post-hoc calibrated using ECE and temperature scaling (Guo et al., 2017). Multicalibration and multiaccuracy strengthen this requirement by asking for calibrated or unbiased predictions over many subgroups or auditing functions (Kleinberg et al., 2016; Hébert-Johnson et al., 2018; Kim et al., 2019). In MoEs the subgroups are latent and overlapping, induced by the router rather than fixed in advance; Appendix C formalizes this connection through routing-weighted auditors $r_k(x)\phi(f(x))$.

**Adversarial reweighting.** Our objective is an instance of adversarial loss reweighting. Distributionally robust optimization learns predictors that perform well under reweightings of the training distribution (Scarf et al., 1957; Delage and Ye, 2010; Subbaswamy et al., 2022); CVaR and related bounded-density objectives emphasize worst-tail examples (Rockafellar et al., 2000; Levy et al., 2020), while entropic or tilted risks emphasize high-loss examples smoothly through exponential tilting (Li et al., 2023). Our adversary is the entropy-balancing member of this family: entropy balancing chooses weights minimally distorted from

a base distribution in relative entropy subject to moment constraints (Hainmueller, 2012), and we apply it adversarially by constraining the model's current proper loss rather than external covariate moments, so the dual exponential tilt is the least-distorted reweighting that stresses high-loss routing-overlap regions without collapsing to a hard subset (§5, Appendix B). Focal loss and variants (Lin et al., 2017; Li et al., 2020; Zhang et al., 2021; Cui et al., 2019) also reweight difficult examples, but without the maximum-entropy or distributional-robustness interpretation, and have been connected to proper-scoring calibration (Mukhoti et al., 2020; Komisarenko and Kull, 2024; Lin et al., 2025).

## 8. Discussion and conclusion

The main lesson is that reliable MoEs require more than reliable experts. Hard routing has a simple expert-confidence bottleneck, so changes in the prevalence of routing regions can preserve calibration. Soft routing removes this bottleneck: the aggregate prediction collapses many routing configurations to the same confidence value, and calibration can fail when the test distribution reweights those configurations. The experiments support this: MoCaE shows that calibrating experts individually does not remove mixture-level miscalibration, while Robust MoE and Robust Filtered improve calibration on shifted or ambiguous subsets across CIFAR-10H, PACS, and CivilComments. These gains come from pressuring the routing-overlap configurations where the failure lives, not from generic high-loss regularization.

A practical implication is that MoE calibration should be evaluated at the aggregate level and under meaningful subpopulation or routing shifts. Average accuracy and per-expert calibration can both look acceptable while final probabilities fail on subsets where the router changes the configuration mix. Post-hoc temperature scaling is a useful complement but not a substitute: a single scalar cannot repair a routing-dependent mismatch between training- and test-time configurations.

The proposed adversarial reweighting is deliberately conservative. High proper loss is only a surrogate for routing-induced calibration error, and some high-loss examples are difficult for reasons unrelated to routing; Robust Filtered addresses this by applying the robust term only where routing appears consequential while keeping an ERM anchor on all examples, which is also why it tends to give a better accuracy–calibration tradeoff than the full reweighting. The main limitations are that we evaluate moderate-size MoEs with four experts, use proxy hard subsets rather than direct interventions on the router, and summarize calibration with ECE, which can hide localized errors. Future work should test larger sparse and generative MoEs, develop more direct router diagnostics, and study guarantees for more structured test-time shifts.

## Acknowledgments

G.W. was partially funded by a discretionary fund at Johns Hopkins University. D.P., S.S., and A.L. were partially funded by the Gordon and Betty Moore Foundation grant #12128. The authors would like to thank anonymous reviewers for helpful feedback.

## Impact statement

Better-calibrated probabilities support more reliable downstream decisions. Mixture-of-experts is a popular framework, so calibration of MoE is relevant for the safety and reliability of many systems. Beyond this, the societal consequences of our work are those typical of advancing machine learning, and we do not feel any further specific concern needs to be highlighted here.

## Code

Code to reproduce experiments and figures are provided at github.com/ginawong/calibrated_moe.

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

# A. Proof of Proposition 4.1

Recall $S = S_{\text{soft}}(X)$, that $f(X)$ is a deterministic function of $S$, and that $m(s) = \mathbb{E}_{P_{\text{train}}}[Y \mid S = s]$. Write $P = P_{\text{train}}$, let $Z = f(X)$, and let $P_a = P_{\text{test}}$ denote the test distribution obtained by the density $a(S)$. Thus $a(S) \geq 0$, $\mathbb{E}_P[a(S)] = 1$, and for any integrable $H$,

$$\mathbb{E}_{P_a}[H(S, Y)] = \mathbb{E}_P[a(S)H(S, Y)].$$

Because the reweighting depends only on $S$, it leaves the conditional law of $Y$ given $S$ unchanged, and hence

$$\mathbb{E}_{P_a}[Y \mid S] = \mathbb{E}_P[Y \mid S] = m(S).$$

We first record the conditional change-of-measure identity used below. Since $Z = f(X)$ is $\sigma(S)$-measurable, the $P_a$-law of $Z$ is absolutely continuous with respect to the $P$-law of $Z$, with Radon–Nikodym derivative $\mathbb{E}_P[a(S) \mid Z]$. Moreover, for any integrable $\varphi(S)$,

$$\mathbb{E}_{P_a}[\varphi(S) \mid Z] = \frac{\mathbb{E}_P[a(S)\varphi(S) \mid Z]}{\mathbb{E}_P[a(S) \mid Z]} \tag{9}$$

on the set where the denominator is positive. To verify this identity, test both sides against an arbitrary bounded measurable function $\psi(Z)$. On one hand,

$$\begin{aligned}
\mathbb{E}_{P_a}[\psi(Z)\varphi(S)] &= \mathbb{E}_P[a(S)\psi(Z)\varphi(S)] \\
&= \mathbb{E}_P[\psi(Z)\mathbb{E}_P[a(S)\varphi(S) \mid Z]].
\end{aligned}$$

On the other hand, using the definition of $P_a$ and conditioning on $Z$,

$$\begin{aligned}
\mathbb{E}_{P_a}\left[\psi(Z)\frac{\mathbb{E}_P[a(S)\varphi(S) \mid Z]}{\mathbb{E}_P[a(S) \mid Z]}\right] &= \mathbb{E}_P\left[a(S)\psi(Z)\frac{\mathbb{E}_P[a(S)\varphi(S) \mid Z]}{\mathbb{E}_P[a(S) \mid Z]}\right] \\
&= \mathbb{E}_P[\psi(Z)\mathbb{E}_P[a(S)\varphi(S) \mid Z]],
\end{aligned}$$

which proves (9).

Applying (9) with $\varphi(S) = m(S) - Z$ gives, for $P_a$-almost every prediction level $p$, equivalently for $P$-almost every prediction level $p$ with $\mathbb{E}_P[a(S) \mid Z = p] > 0$,

$$\begin{aligned}
\mathbb{E}_{P_a}[Y - f(X) \mid f(X) = p] &= \mathbb{E}_{P_a}[m(S) - Z \mid Z = p] \\
&= \frac{\mathbb{E}_P[a(S)(m(S) - p) \mid Z = p]}{\mathbb{E}_P[a(S) \mid Z = p]} \\
&= \frac{\mathbb{E}_{P_{\text{train}}}[a(S)(m(S) - p) \mid f(X) = p]}{\mathbb{E}_{P_{\text{train}}}[a(S) \mid f(X) = p]}.
\end{aligned}$$

The first equality uses the tower property and $\mathbb{E}_{P_a}[Y \mid S] = m(S)$; the second is the conditional change of measure for the regular conditional law of $S$ given the prediction level. This proves (4).

It remains to prove the equivalence. Let $h(S) = m(S) - f(X)$. If $h(S) = 0$ $P$-almost surely, then $h(S) = 0$ $P_a$-almost surely for every admissible reweighting $a$, since $P_a$ is absolutely continuous with respect to $P$. Therefore

$$\mathbb{E}_{P_a}[Y - f(X) \mid f(X)] = \mathbb{E}_{P_a}[h(S) \mid f(X)] = 0$$

for every admissible $a$, so aggregate calibration is preserved.

Conversely, suppose $h(S) \neq 0$ with positive $P$-probability. Then either $B_+ = \{h(S) > 0\}$ or $B_- = \{h(S) < 0\}$ has positive $P$-probability. Assume $P(B_+) > 0$; the negative case is symmetric. Since $h(S)$ is $\sigma(S)$-measurable, $B_+$ is a routing-configuration event, so

$$a(S) = \frac{\mathbb{1}_{B_+}}{P(B_+)}$$

is an admissible configuration reweighting. Under the corresponding test distribution $P_a$, we have $h(S) > 0$ almost surely. Hence $\mathbb{E}_{P_a}[h(S) \mid f(X)] \geq 0$ almost surely, and this conditional expectation cannot vanish on any set of prediction levels

with positive $P_a$-measure: if it did, the integral of the nonnegative random variable $h(S)$ over the preimage of that set would be zero, contradicting $h(S) > 0$ almost surely there. Therefore

$$\mathbb{E}_{P_a}[Y - f(X) \mid f(X)] = \mathbb{E}_{P_a}[h(S) \mid f(X)] > 0$$

on a set of prediction levels with positive $P_a$-measure, so aggregate calibration fails under this admissible reweighting. If instead $P(B_-) > 0$, the same argument with $a(S) = \mathbf{1}_{B_-}/P(B_-)$ gives a strictly negative conditional residual on a set of positive measure. Thus aggregate calibration is preserved under all admissible configuration reweightings if and only if $m(S) = f(X) \, P_{\text{train}}$-almost surely. The final equivalent formulation follows by conditioning this almost-sure identity on the prediction level $f(X) = p$.

# B. Variants of adversarial reweighting

## B.1. Entropy-balanced adversarial reweighting

On a minibatch with losses $L_i = L_\theta(x_i, y_i)$, our adversary chooses a distribution $q \in \Delta_n$ over examples. The goal is to stress the examples on which the aggregate predictor currently incurs large proper loss, while changing the empirical distribution as little as possible. This is the entropy-balancing criterion: among reweightings that raise the expected loss to an adversarial level, choose the one closest to the empirical weights in relative entropy. Routing posterior shift acts on diffuse overlap regions rather than a few discrete groups, so we want this smooth, least-distorted adversary rather than a sparse one.

Formally, for any attainable target loss level $c$ above the empirical mean,

$$q_c^\star \in \arg\min_{q \in \Delta_n} \mathrm{KL}(q \,\|\, u) \qquad \text{subject to} \qquad \sum_{i=1}^n q_i L_i = c, \qquad u_i = \tfrac{1}{n}. \tag{10}$$

Classical entropy balancing matches externally specified covariate moments (Hainmueller, 2012); here the moment is the model's own current proper loss, so the adversary builds the least-distorted reweighting that exposes the predictor's high-loss examples. The Lagrangian for (10) assigns a multiplier $\eta \geq 0$ to the loss constraint, and its solution is the exponential tilt

$$q_i^\eta = \frac{\exp(\eta L_i)}{\sum_{j=1}^n \exp(\eta L_j)}. \tag{11}$$

We treat $\eta$ as a hyperparameter controlling the adversary's strength: $\eta = 0$ recovers the uniform distribution, and larger $\eta$ places more mass on higher-loss examples.

We train on the expected proper loss under this entropy-balanced adversary:

$$\mathcal{L}_{\text{R-MoE}}(\theta) = \sum_{i=1}^n q_i^\eta L_i, \qquad L_i := L_\theta(x_i, y_i). \tag{12}$$

This is the objective used in our implementation. The weights $q_i^\eta$ are recomputed from the current losses at every step, so gradients flow through both $L$ and $q^\eta$.

**KL decomposition.** Substituting $q_i^\eta = e^{\eta L_i}/Z$, with $Z = \sum_j e^{\eta L_j}$, into the KL divergence to the uniform $u_i = 1/n$,

$$\tfrac{1}{\eta} \mathrm{KL}(q^\eta \| u) = \tfrac{1}{\eta} \sum_i q_i^\eta \big[\log(n \, q_i^\eta)\big] = \tfrac{1}{\eta} \sum_i q_i^\eta \big[\eta L_i - \log Z + \log n\big]$$

$$= \sum_i q_i^\eta L_i - \tfrac{1}{\eta} \log \tfrac{Z}{n} = \mathcal{L}_{\text{R-MoE}}(\theta) - \rho_\eta(L),$$

where $\rho_\eta(L)$ is the standard entropic risk,

$$\rho_\eta(L) := \frac{1}{\eta} \log \left( \frac{1}{n} \sum_{i=1}^n e^{\eta L_i} \right). \tag{13}$$

We drop the argument and write $\rho_\eta$ when the loss vector is understood from context. Rearranging gives the identity used in the body,

$$\mathcal{L}_{\text{R-MoE}}(\theta) = \rho_\eta + \tfrac{1}{\eta} \mathrm{KL}(q^\eta \| u). \tag{14}$$

Since $\mathrm{KL} \geq 0$, $\mathcal{L}_{\text{R-MoE}} \geq \rho_\eta$ pointwise, with equality only when $q^\eta = u$, i.e. when all minibatch losses are equal.

**Upper-bound surrogate for KL-ball DRO.** For the KL-ball uncertainty set $\{q : \mathrm{KL}(q\|u) \leq \varepsilon\}$, the standard dual bound gives, for any $\eta > 0$,

$$\sup_{q \,:\, \mathrm{KL}(q\|u) \leq \varepsilon} \sum_i q_i L_i \leq \rho_\eta + \tfrac{\varepsilon}{\eta} \leq \mathcal{L}_{\text{R-MoE}} + \tfrac{\varepsilon}{\eta}. \tag{15}$$

Thus $\mathcal{L}_{\text{R-MoE}}$ is a conservative upper-bound surrogate for the KL-ball DRO objective, looser than the entropic risk by exactly the induced KL term in (14). The looseness is *induced* rather than imposed: it grows whenever the current losses concentrate $q^\eta$ on a small set, which is the regime §4 identifies as vulnerable.

**Gradient: the covariance identity.** Differentiating (12) through both $L$ and $q^\eta$,

$$\nabla_\theta \mathcal{L}_{\text{R-MoE}} = \sum_i q_i^\eta \nabla_\theta L_i \; + \; \sum_i L_i \nabla_\theta q_i^\eta.$$

The first sum is the entropic-risk gradient (by the envelope theorem, or directly: $\nabla_\theta \rho_\eta = \sum_i q_i^\eta \nabla_\theta L_i$). Using $\nabla_\theta q_i^\eta = \eta\, q_i^\eta\big(\nabla_\theta L_i - \sum_j q_j^\eta \nabla_\theta L_j\big)$, the second sum collapses to

$$\sum_i L_i \nabla_\theta q_i^\eta \; = \; \eta\, \text{Cov}_{q^\eta}\big(L, \nabla_\theta L\big).$$

This is precisely $\nabla_\theta\big[\frac{1}{\eta}\text{KL}(q^\eta\|u)\big]$, as (14) requires, so

$$\nabla_\theta \mathcal{L}_{\text{R-MoE}} = \underbrace{\nabla_\theta \rho_\eta}_{\text{envelope-theorem term}} + \underbrace{\eta\, \text{Cov}_{q^\eta}\big(L, \nabla_\theta L\big)}_{\nabla_\theta\big[\frac{1}{\eta}\text{KL}(q^\eta\|u)\big]}. \tag{16}$$

The implementation backpropagates through $q^\eta$ and so computes exactly (16); training on $\mathcal{L}_{\text{R-MoE}}$ is therefore not the same as training on $\rho_\eta$, even though both are built from the same exponential-tilt weights.

**Relation to tilted ERM.** The classical tilted/entropic risk $\rho_\eta$ has gradient weights $q_i^\eta$ by the envelope theorem (Li et al., 2023); for a fixed loss vector $L$ one also has $\mathcal{L}_{\text{R-MoE}} = \partial_\eta[\eta\, \rho_\eta(L)]$. These identities explain the close kinship between our weights and tilted ERM, but, by (16), they do not identify $\mathcal{L}_{\text{R-MoE}}$ and $\rho_\eta$ as objectives in $\theta$: the covariance term is present in one and absent in the other.

**Why entropy balancing for routing posterior shift.** Routing posterior shift is rarely observed through explicit group labels. It appears through changes in the prevalence of latent contexts and through shifts in the regions where the router is uncertain or experts disagree. A hard subset adversary can locate rare high-loss pockets, but it may also overfit to discontinuous minibatch artifacts. The entropy-balanced adversary is more conservative: it uses the current loss to find the direction of stress, but otherwise stays as close as possible to the empirical distribution. This keeps the robust objective sensitive to the brittle routing-overlap regions of §4 while preserving smooth gradients for the router and experts.

### B.2. Alternative reweighting geometries

Two related objectives are worth contrasting with the entropy-balanced reweighting used above.

**KL-penalized adversary (entropic DRO).** The same exponential weights arise from the KL-*penalized* adversary

$$\max_{q\in\Delta_n}\left\{\sum_{i=1}^n q_i L_i - \tfrac{1}{\eta}\text{KL}(q\,\|\,u)\right\}, \tag{17}$$

the usual route to KL-ball DRO bounds. This adversary and entropy balancing are Lagrangian duals—one fixes the penalty $1/\eta$, the other fixes the loss level $c$—and both return the weights in (11). They differ in the training objective they induce, not in the adversary: minimizing against the penalized adversary gives the entropic risk $\rho_\eta$, whereas we train on the expected loss under those same weights, $\mathcal{L}_{\text{R-MoE}} = \rho_\eta + \frac{1}{\eta}\text{KL}(q^\eta\|u)$, which adds the induced concentration term. The penalized form has the cleaner certificate—$\rho_\eta + \varepsilon/\eta$ bounds the KL-ball worst case—and is the natural starting point when the design knob is the radius itself. We lead with the constrained form because its constraint is on the loss moment, the quantity the multiaccuracy bound in Appendix C ties to residual calibration moments.

**CVaR (bounded density ratio).** A genuinely different geometry replaces the entropy penalty with a hard density-ratio bound. Define

$$\mathcal{W}_\Gamma := \{w : \mathcal{X} \times \mathcal{Y} \to [0,\Gamma] \mid \mathbb{E}_{P_{\text{train}}}[w(X,Y)] = 1\},$$

giving the robust score

$$\min_\theta \sup_{w\in\mathcal{W}_\Gamma} \mathbb{E}_{P_{\text{train}}}[w(X,Y) L_\theta(X,Y)]. \tag{18}$$

The adversary can place weight $\Gamma$ on at most a $1/\Gamma$ fraction of the population, so the inner supremum is the worst-tail (CVaR) risk of the loss distribution (Rockafellar et al., 2000); in finite samples it corresponds to averaging roughly the largest $m \approx n/\Gamma$ losses in the batch, up to a fractional weight at the boundary. Unlike the entropic geometries above, this changes the weights themselves, not just the objective.

CVaR gives the sharpest distributional guarantee of the three: under a bounded density-ratio shift, (18) directly upper-bounds the test loss. Its drawback for our setting is the hard top-$m$ cutoff, which concentrates the gradient signal on a few examples per batch—mismatched to the diffuse routing-overlap regions of §4—and makes the objective non-smooth. We therefore use CVaR as a theoretical reference rather than a training objective. A useful bridge nonetheless connects it to the geometry we use: any $w \in \mathcal{W}_\Gamma$ satisfies $\mathrm{KL}(P_w \| P_{\text{train}}) \leq \log \Gamma$, and a standard variational bound implies that for any $\eta > 0$,

$$\sup_{w \in \mathcal{W}_\Gamma} \mathbb{E}_{P_{\text{train}}}[w L_\theta] \leq \tfrac{1}{\eta} \log \Gamma + \tfrac{1}{\eta} \log \mathbb{E}_{P_{\text{train}}}[\exp(\eta L_\theta)] . \tag{19}$$

The entropic risk is thus a smooth upper-bound surrogate, up to an additive constant, for the box-constrained CVaR risk; by (14), $\mathcal{L}_{\text{R-MoE}} + (1/\eta) \log \Gamma$ is likewise an upper-bound surrogate.

## C. Multiaccuracy interpretation of the robust objective

The residual-moment connection is easiest to state using the squared residual. Let $\mathcal{W}$ be any family of nonnegative normalized weights, and define

$$J_{\mathcal{W}}(f) := \sup_{w \in \mathcal{W}} \mathbb{E}_{P_{\text{train}}}\left[w(X, Y)(Y - f(X))^2\right].$$

Fix any bounded class $\Phi$ of functions $\phi : [0, 1] \to [0, 1]$. For each expert $k$ and $\phi \in \Phi$, define the routing-induced auditor

$$g_{k,\phi}(x) := r_k(x) \, \phi(f(x)) \in [0, 1].$$

Then, for all $k$ and all $\phi \in \Phi$,

$$\sup_{w \in \mathcal{W}} |\mathbb{E}_{P_{\text{train}}}[w(X, Y) \, g_{k,\phi}(X) \, (Y - f(X))]| \leq \sqrt{J_{\mathcal{W}}(f)}.$$

This follows from Cauchy–Schwarz: since $0 \leq g_{k,\phi}(x) \leq 1$ and $\mathbb{E}_{P_{\text{train}}}[w] = 1$, we have $\mathbb{E}_{P_{\text{train}}}[w \, g_{k,\phi}(X)^2] \leq 1$. Consequently, if the worst-case squared residual is small over a reweighting family, then the residual cannot systematically align with any auditor formed from the routing weights, connecting the objective to multiaccuracy (Kim et al., 2019).

For binary labels and $p \in (0, 1)$,

$$(y - p)^2 \leq -y \log p - (1 - y) \log(1 - p) = \ell_{\text{CE}}(p, y).$$

Thus, a robust cross-entropy bound over the same reweighting family also upper-bounds $J_{\mathcal{W}}(f)$.

For the proposed objectives, the relevant choices of $\mathcal{W}$ are the reweightings used during training. For Robust MoE, this is the full-minibatch entropic weighting $q^\eta$; for Robust Filtered, the bound applies separately to the uniform ERM term over the full minibatch and to the entropic weighting $q_A^\eta$ over the routing-relevant set $A$. Thus, each training term controls the corresponding weighted residual moments in the multiaccuracy bound.

The value-level certificate (8) makes this concrete for Robust MoE over a KL ball. Let

$$\mathcal{W}_\varepsilon := \{w \geq 0 : \, \mathbb{E}_{P_{\text{train}}}[w] = 1, \, \text{KL}(P_w \| P_{\text{train}}) \leq \varepsilon\}.$$

For cross-entropy, or any loss that pointwise dominates the squared residual (with equality for the Brier score), the inequality $(Y - f(X))^2 \leq L_\theta(X, Y)$ holds, and combining it with the certificate (8) gives, for every $w \in \mathcal{W}_\varepsilon$,

$$\mathbb{E}_{P_w}\left[(Y - f(X))^2\right] \leq \mathbb{E}_{P_w}[L_\theta] \leq \mathcal{L}_{\text{R-MoE}}^{\text{pop}}(\theta) + \tfrac{\varepsilon}{\eta}.$$

Taking the supremum over $w \in \mathcal{W}_\varepsilon$ yields $J_{\mathcal{W}_\varepsilon}(f) \leq \mathcal{L}_{\text{R-MoE}}^{\text{pop}}(\theta) + \varepsilon/\eta$, so by the Cauchy-Schwarz bound above, every routing-weighted auditor $g_{k,\phi}(x) = r_k(x)\phi(f(x))$ satisfies

$$\sup_{w \in \mathcal{W}_\varepsilon} |\mathbb{E}_{P_{\text{train}}}[w \, g_{k,\phi}(X) \, (Y - f(X))]| \leq \sqrt{\mathcal{L}_{\text{R-MoE}}^{\text{pop}}(\theta) + \tfrac{\varepsilon}{\eta}}.$$

Controlling the robust loss therefore controls residual alignment with routing-weighted auditors over a KL-controlled reweighting family. This is a statement about robust-risk values, not about individual gradient steps: as Appendix B shows, the implementation differentiates through the tilt weights $q^\eta$. For Robust Filtered the same reasoning applies to the filtered robust term after restricting to the routing-relevant set $A$, while the ERM anchor controls the unfiltered residual moment under the empirical training distribution.

**From multiaccuracy to multicalibration.** This bound applies to unnormalized moments, i.e., multiaccuracy, rather than the conditional expectations required for multicalibration (Hébert-Johnson et al., 2018). To convert it to a conditional calibration guarantee such as $|\mathbb{E}_{P_w}[Y - f(X) \mid g_{k,\phi}(X) = 1]|$, one additionally needs a lower bound on the shifted mass $\mathbb{E}_{P_{\text{train}}}[w(X, Y) \, g_{k,\phi}(X)]$ for the slice of interest.

# D. Additional experimental details

## D.1. Baselines

**Frequency-aware Gradient Rectification (FGR) (Zhang et al., 2025)**   FGR is a training procedure designed to improve calibration under covariate shift for single models. Each training step computes two gradients: one from the main classification loss on a frequency-perturbed version of the batch, and one from a differentiable calibration loss (soft ECE (Karandikar et al., 2021)) on the original batch. If these two gradients conflict—i.e., their dot product is negative, meaning improving classification would worsen calibration—then the main gradient is projected onto the hyperplane orthogonal to the calibration gradient, removing the conflicting component. When the gradients agree, the main gradient is used as-is.

The frequency perturbation works by applying JPEG-style $8 \times 8$ block Discrete Cosine Transform (DCT) filtering to a small random fraction ($\nu = 0.05$) of training images. Images are converted to YCbCr color space, each $8 \times 8$ block is transformed, quantized using scaled standard JPEG quantization matrices, and reconstructed via inverse DCT. The quantization strength is randomly sampled from $\lambda \in \{15, 18, 25\}$ per image, simulating mild frequency-domain corruptions. This is meant to make the model robust to the kind of high-frequency perturbations that arise in real-world covariate shift scenarios.

Matching the paper, we use cross-entropy as the main loss and soft ECE as the calibration loss, and we use vanilla warmup for 20 epochs before switching to gradient rectification. Each batch requires two forward passes and two backward passes (one per objective), plus CPU-based DCT filtering, making training roughly 2.5x slower per epoch than vanilla.

FGR is a training procedure, so combining it with our robust method requires deciding which method provides the main loss for the rectification framework. We combine FGR with Robust MoE by substituting our entropy-balanced robust loss for cross-entropy as the main loss: the main gradient is computed from the full entropy-balanced objective on the frequency-perturbed batch, while the calibration gradient (soft ECE on the original batch) and the rectification logic are unchanged. We note that the motivation for this combination is weaker, as FGR was designed for covariate shift on single models, not MoE routing fragility, and its DCT-based frequency filtering targets the kind of high-frequency perturbations that arise from image corruptions, which have essentially no connection to the subpopulation structure that drives the routing-induced shift.

**MoCaE (Oksuz et al., 2024; Roschewitz et al., 2025)**   This paper is not a traditional baseline in the sense that they propose a method to compare against; instead, the authors argue that for single models under distribution shift, post-hoc calibration with a small amount of semantic out-of-distribution (OOD) data is often more effective than more complex methods that use OOD data during calibration. For ensembles, the authors suggest that OOD data is unnecessary for maintaining calibration under shift, and that the best approach is to calibrate before ensembling, rather than calibrating the final averaged predictor afterward.

For our setting, the most relevant part of this paper is the result on calibrating ensembles. Ensembles are not the same thing as MoE, but if we were to draw an analogy between the set of ensemble members and the set of MoE experts, then the suggested method for MoE would be to calibrate the experts before they are combined. This is essentially our MoCaE baseline, which we already include in our comparisons.

We additionally report results with post-hoc temperature scaling, where the temperature selected on a held-out validation set (we use 5,000 images for validation, and the remaining 45,000 images for training). (This allows us to separate improvements in calibration due to training from those achievable with post-hoc calibration alone.)

## D.2. Datasets and models

**CIFAR-10H**   CIFAR-10H is the CIFAR-10 dataset augmented with human soft labels collected from $\approx 50$ annotators per test image (Peterson et al., 2019). We define images with low human agreement (max annotation probability $< 0.7$) as *hard*. These ambiguous images make up $\approx 3.3\%$ of the test set (327 of 10,000 images). All methods are trained on the standard CIFAR-10 training set (50,000 images), with 10% (5,000 images) withheld for validation and temperature scaling.

The model uses a ResNet-18 backbone adapted for $32 \times 32$ CIFAR images ($3 \times 3$ initial convolution, no max pooling, three residual blocks producing a 256-dimensional representation). The 256-dimensional output is fed into the MoE classification head, which consists of 4 expert linear classifiers and a 2-layer MLP router. The entire model is trained end-to-end using AdamW (lr=1e-3, weight decay=1e-4) with cosine annealing over 50 epochs. For the robust methods, a 20-epoch warmup period trains with standard cross-entropy before the robust objective is applied.

**PACS Dataset (4 domains, 7 classes)**

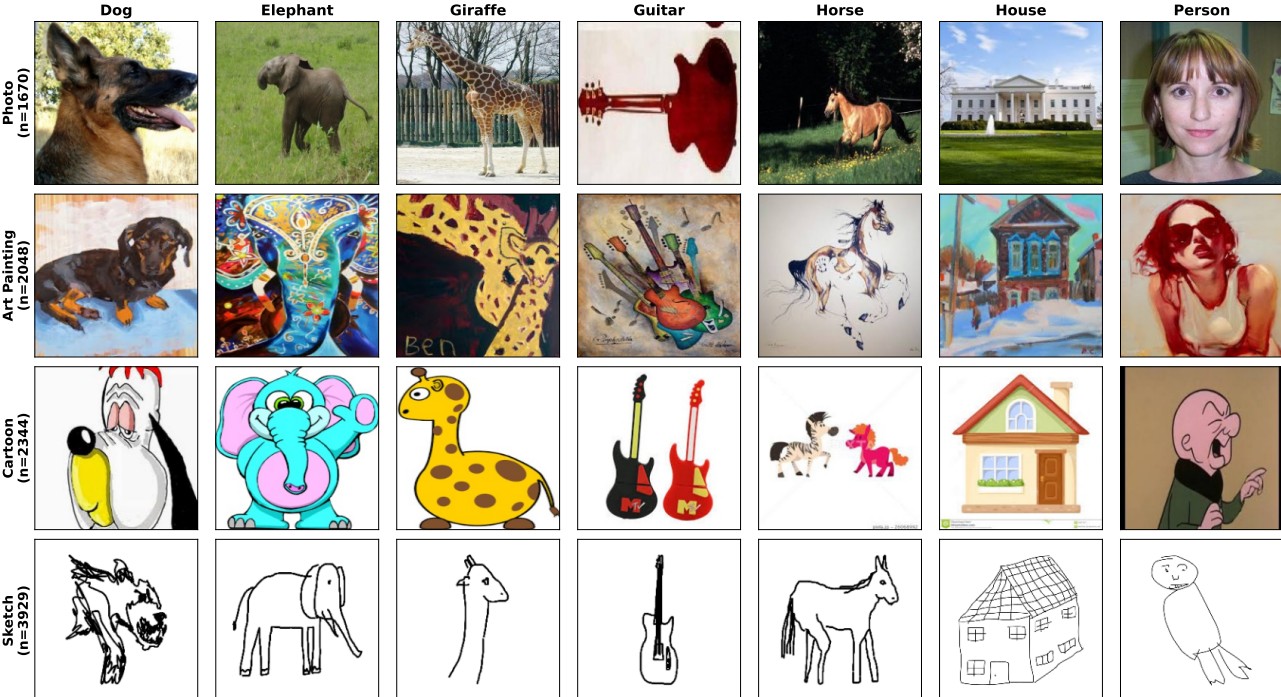

*Figure D.1.* Example images from the PACS dataset across four domains.

**PACS**   PACS is a widely used benchmark for evaluating image models under domain shift (Li et al., 2017). It contains object categories from four domains: Photo, Art Painting, Cartoon, and Sketch (Figure D.1). Because these domains have visually distinct styles, evaluation on PACS is useful for studying whether model behavior remains stable when the image style changes.

PACS contains approximately 10,000 images of 7 object categories (dog, elephant, giraffe, guitar, horse, house, person) across four visually distinct domains. Following the standard leave-one-domain-out protocol, the model trains on three source domains and is evaluated on the held-out target domain. For our evaluation, Photo, Art Painting, Cartoon, and Sketch all take turns as the target domain. 10% of the training data is withheld for validation and temperature scaling.

The model uses a ResNet-18 backbone pretrained on ImageNet as the shared feature extractor. The 512-dimensional output is fed into the MoE classification head, which consists of 4 expert classifiers and a 2-layer MLP router. The entire model, backbone and MoE head, is fine-tuned end-to-end using AdamW with a learning rate of 1e-4 and cosine annealing over 50 epochs.

**CivilComments**   CivilComments is a text classification benchmark from WILDS (Koh et al., 2021), where the goal is to predict whether an online comment is toxic under a natural subpopulation shift setting. This experiment extends our approach to a transformer-based architecture operating on text, demonstrating that the robust training framework is not limited to convolutional backbones or vision tasks.

The WILDS CivilComments dataset contains approximately 270,000 training comments and 134,000 test comments. For tractable end-to-end fine-tuning of the DistilBERT backbone, we randomly subsample the training comments to 50000 (fixed seed); from this subsample, we withhold the last 10% (5000 comments) as a validation split used solely for temperature scaling, leaving 45000 comments for training. The full WILDS test split is used for evaluation. Of the test set, roughly 41% (55,000 comments) mention at least one identity group. These comments are considered hard, as toxicity classifiers generally have high false positive rates on even benign mentions of minority groups. This is a much larger hard subpopulation than CIFAR-10H, where only 3% of the test set is hard. The identity group annotations are used solely at evaluation time to define the hard subset.

The model uses a DistilBERT (Sanh et al., 2019) backbone (66M parameters, pretrained on English text) as the feature extractor. Each comment is tokenized and passed through the full transformer encoder; the output (768-dimensional [CLS] representation) is then fed into our MoE classification head, where the router and expert networks operate on these learned text representations to produce the final toxicity prediction. The entire model, including the backbone and MoE head, is fine-tuned end-to-end.

### D.3. Implementation of the proposed methods

The body of the paper writes the proposed objectives in terms of a generic strictly proper loss $L_\theta(X, Y)$. In all experiments we instantiate this loss as multiclass cross-entropy applied to the aggregate mixture probabilities,

$$L_\theta(x, y) = -\log f_\theta(x)_y = -\log \sum_{k=1}^{K} r_k(x) f_k(x)_y,$$

where $f_k(x) \in \Delta^{C-1}$ is each expert's softmax distribution over the $C$ classes and $r(x) \in \Delta^{K-1}$ is the routing distribution. Cross-entropy is strictly proper, so the multiaccuracy bound in Appendix C applies via the pointwise inequality $(y - p)^2 \leq \ell_{\mathrm{CE}}(p, y)$.

**Entropy-balanced tilted-softmax objective.** Both proposed methods evaluate the maximum-entropy adversarial weights from (11) on each minibatch. Concretely, given per-example losses $L_1, \ldots, L_n$ on a minibatch of size $n$, we form

$$q_i^\eta = \frac{\exp(\eta L_i)}{\sum_{j=1}^{n} \exp(\eta L_j)}, \qquad \mathcal{L}(\theta) = \sum_{i=1}^{n} q_i^\eta L_i,$$

and backpropagate through the scalar objective as written, so gradients flow through both the losses $L_i$ and the weights $q_i^\eta$. Thus the update is the full gradient of the entropy-balanced objective, including the covariance term that implements the induced-concentration penalty derived in Appendix B.1.

Appendix B.2 relates this diffuse exponential-tilt geometry to the sparse top-tail geometry of CVaR, and the next subsection (§D.6) reports the empirical correspondence between $\eta$ and the effective density-ratio bound $\Gamma$.

**Warmup phase.** Adversarial reweighting at initialization is uninformative: before the model has learned useful features, the largest losses are dominated by random initialization rather than by routing fragility. We therefore train every robust method with a warmup phase that uses standard ERM on the aggregate cross-entropy loss, $\mathcal{L}_{\mathrm{warm}}(\theta) = \frac{1}{n} \sum_i L_i$, for the first $T_{\mathrm{warm}}$ epochs of training. After the warmup, the optimizer switches to the robust objective for the remaining epochs. The cosine learning-rate schedule runs over the full training horizon and is unaffected by the warmup-to-robust transition. Default values are $T_{\mathrm{warm}} = 20$ for the 50-epoch image experiments and $T_{\mathrm{warm}} = 2$ for the 5-epoch CivilComments fine-tuning run; both values correspond to 40% of the training horizon, so warmup occupies a comparable fraction of training across datasets.

**Robust MoE.** Robust MoE applies the tilted-softmax weights to the full minibatch:

$$\mathcal{L}_{\text{R-MoE}}(\theta) = \sum_{i=1}^{n} q_i^\eta L_i \quad \text{with} \quad L_i = -\log f_\theta(x_i)_{y_i}.$$

Every minibatch sample enters the forward objective with positive tilted weight $q_i^\eta$, monotone in its current loss. We use $\eta = 2.0$ for all reported Robust MoE results.

**Robust Filtered.** Robust Filtered restricts the entropic reweighting to a routing-relevant subset $A \subseteq \{1, \ldots, n\}$ and adds an ERM anchor over the full minibatch:

$$\mathcal{L}_{\text{RF-MoE}}(\theta) = \frac{1}{n} \sum_{i=1}^{n} L_i + \sum_{i \in A} q_{i,A}^\eta L_i, \qquad q_{i,A}^\eta = \frac{\exp(\eta L_i)}{\sum_{j \in A} \exp(\eta L_j)}.$$

A sample is placed in $A$ if either of two routing-relevance conditions holds:

1. **Mixture regret.** Let $L_i^{\text{mix}} = -\log f_\theta(x_i)_{y_i}$ and $L_i^{\text{best}} = \min_k -\log f_k(x_i)_{y_i}$. The regret $R_i = (L_i^{\text{mix}} - L_i^{\text{best}})_+$ is positive whenever the mixture prediction is strictly worse than the best individual expert. We mark $i$ as routing-relevant if $R_i > \tau_{\text{regret}}$, with $\tau_{\text{regret}} = 10^{-6}$ in all experiments.

2. **Routing-weighted disagreement.** We compute $d_i = \sum_{k=1}^{K} r_k(x_i) \|f_k(x_i) - f_\theta(x_i)\|_2^2$, the routing-weighted variance of expert probability vectors around the mixture prediction. We mark $i$ as routing-relevant if $d_i > \tau_{\text{disagree}}$, with $\tau_{\text{disagree}} = 0.01$.

A sample joins $A$ as soon as either criterion fires. The first criterion uses the actual labeled regret of the mixture against the best expert and is informative only when training labels are available; the second criterion uses only model outputs and fires whenever the experts disagree enough that the routing weights materially affect the prediction. Each criterion alone is too narrow—some routing-sensitive examples are confidently miscalibrated even when a single expert is best, and some high-regret examples have only one disagreeing expert—so we take the union. As with Robust MoE, we use $\eta = 2.0$ for all reported Robust Filtered results.

**Why an ERM anchor is needed.**  The shape parameter $\eta$ alone cannot decouple emphasis on routing-relevant tail samples from emphasis on intrinsically hard samples. Lowering $\eta$ recovers gradient signal on routine-easy samples but also reduces emphasis on routing-relevant tail samples; raising $\eta$ does the reverse. The ERM anchor in $\mathcal{L}_{\text{RF-MoE}}$ keeps a uniform training signal over the full minibatch, while the robust term can operate at higher $\eta$ on routing-relevant samples. This decouples broad accuracy improvement from targeted robustness pressure, without relying on $\eta$ alone to balance the two.

## D.4. Training hyperparameters

All models share the same MoE architecture: a shared backbone produces a feature representation that is consumed by $K{=}4$ linear expert heads and a 2-layer MLP router (hidden width 128, softmax output). The entire model—backbone, experts, and router—is trained end-to-end with the AdamW optimizer and a cosine annealing learning-rate schedule that runs over the full training horizon. Weight decay is fixed at $10^{-4}$ across all datasets and methods. Dataset-specific hyperparameters are summarized in Table 4.

*Table 4.* Dataset-specific training hyperparameters. All datasets use AdamW with weight decay $10^{-4}$, $K{=}4$ experts, a 2-layer MLP router with hidden width 128, and cosine annealing of the learning rate over the full training horizon. *warmup* is the number of ERM epochs before the robust objective is applied; *init.* is the backbone initialization (ImageNet-pretrained or trained from scratch).

| Dataset | Backbone | Init. | Resolution | Batch | Epochs | Warmup | Learning rate | Augmentation |
|---|---|---|---|---|---|---|---|---|
| CIFAR-10H | ResNet-18 | scratch | 32×32 | 128 | 50 | 20 | $10^{-3}$ | crop, flip |
| PACS | ResNet-18 | ImageNet | 224×224 | 128 | 50 | 20 | $10^{-4}$ | crop, flip |
| CivilComments | DistilBERT | pretrained | 128 tokens | 32 | 5 | 2 | $2\times10^{-5}$ | — |

**Backbone variants.**  The CIFAR-10H ResNet-18 is the standard CIFAR variant: the first convolution is replaced by a 3×3, stride-1 layer with no max-pool, and we use the first three residual stages, yielding a 256-dim feature. The PACS ResNet-18 uses the standard ImageNet stem (7×7, stride 2, max-pool) and all four residual stages, yielding a 512-dim feature. The CivilComments backbone is the public `distilbert-base-uncased` model (66M parameters) and we take the 768-dim `[CLS]` embedding as the feature.

**Data augmentation.**  For both image datasets we apply random horizontal flip and random crop with zero padding (4 pixels for 32×32 inputs, 28 pixels for 224×224 inputs), followed by per-channel mean/standard deviation normalization (CIFAR statistics for CIFAR-10H, ImageNet statistics for PACS). The test transform applies only the normalization. CivilComments inputs are tokenized once at preprocessing time using the DistilBERT tokenizer with maximum length 128 and padded to that length; no additional augmentation is applied.

**Train/validation/test splits.**  For CIFAR-10H, 10% of the 50,000 CIFAR-10 training images (5,000 images) are held out as a validation split that is used only for temperature scaling. For PACS, the leave-one-domain-out target domain is held out as test, and 10% of the source-domain training data is held out as a validation split for temperature scaling. For CivilComments, we subsample the WILDS training split to $50000$ comments (fixed-seed random subsample) and withhold

the last 10% (5000) as a validation split for temperature scaling; the full WILDS test split is used for evaluation, and the WILDS validation split is not used.

**Seeds and reporting.** All reported numbers are mean $\pm$ standard error of the mean across five random seeds, $\{42, 43, 44, 45, 46\}$, applied to model initialization, data shuffling, and (for the robust methods) the warmup-to-robust transition. The same seed set is used across methods to allow seed-paired comparisons.

### D.5. Evaluation protocol

**Top-class ECE.** We report the standard top-class Expected Calibration Error (Guo et al., 2017): predictions are partitioned into $B = 15$ equal-width bins on the predicted-class confidence $\max_c f_\theta(x)_c$, and

$$\mathrm{ECE} \; = \; \sum_{b=1}^{B} \frac{|\mathcal{B}_b|}{N} \, |\mathrm{acc}(\mathcal{B}_b) - \mathrm{conf}(\mathcal{B}_b)| \,,$$

where $\mathcal{B}_b$ is the $b$-th confidence bin, $\mathrm{conf}(\mathcal{B}_b)$ is the mean top-class probability in the bin, and $\mathrm{acc}(\mathcal{B}_b)$ is the empirical top-1 accuracy in the bin. We use 15 bins for all datasets; the same binning is applied to overall ECE and to ECE on hard subsets, restricting the sum to test points in the subset before binning.

**Hard subsets.** For each dataset, the body reports a Hard ECE summary on the test points most directly stressed by routing-induced reweighting. The exact hard-subset definitions are:

- **CIFAR-10H.** Test images with maximum CIFAR-10H human-annotation probability below $0.7$. This corresponds to images on which annotators systematically disagree and is taken as a proxy for routing-overlap regions where multiple experts may have a plausible interpretation. The hard subset contains 327 of the 10000 test images ($\approx 3.3\%$).

- **PACS.** The full held-out target domain is the hard subset; each row of the leave-one-domain-out table reports overall accuracy and ECE on the entire target domain, since the natural shift is the domain itself.

- **CivilComments.** Test comments whose WILDS metadata flags a mention of any of eight identity subgroups (Male, Female, LGBTQ, Christian, Muslim, Other Religions, Black, White). This subset contains $\approx 41\%$ of the test set and isolates the well-documented false-positive failure mode of toxicity classifiers on benign demographic mentions.

The hard-subset notion in CIFAR-10H is the only one of the three that is not directly observable at training time, since human-agreement annotations exist only for the test split.

**Aggregate temperature scaling.** The ECE+TS columns in all tables apply post-hoc temperature scaling to the aggregate mixture probabilities. We collect the aggregate logits $z(x) = \log f_\theta(x)$ on the validation split, fit a single scalar temperature $T > 0$ by minimizing the validation cross-entropy with LBFGS (50 iterations, learning rate 0.01), and rescale the test predictions as $f_\theta^T(x) = \mathrm{softmax}(z(x)/T)$ before computing ECE. The temperature is selected once per (method, seed) pair on the validation split and held fixed for all test evaluations of that pair, including the hard-subset evaluation. For PACS we use the held-out source-domain validation split (§D.4); the target domain is never used for temperature selection.

**Accuracy.** Top-1 accuracy is reported on the same test populations as ECE; *Hard Accuracy* on CIFAR-10H and CivilComments uses the hard subsets defined above.

### D.6. Sensitivity to the robustness parameter $\eta$

The body objectives are stated in terms of the maximum-entropy parameter $\eta$, while Appendix B.2 formulates the robust score in terms of a density-ratio bound $\Gamma$. The two parameters are not equal in general: $\eta$ is the temperature of a softmax over batch losses, and $\Gamma$ is a bound on the worst-case adversarial reweighting. To make the body objective interpretable in the language of the CVaR formulation, we report the empirical correspondence between $\eta$ and an effective density-ratio bound $\Gamma_{\mathrm{eff}}$.

For each batch, the tilted-softmax weights $q^\eta \in \Delta_n$ have an effective sample size given by their perplexity $\mathrm{ppl}(q^\eta) = \exp(-\sum_i q_i^\eta \log q_i^\eta)$, the exponentiated entropy of the weight distribution. Letting $\Gamma_{\mathrm{eff}} = n/\mathrm{ppl}(q^\eta)$ gives the effective density-ratio bound: it is the ratio of the maximum-entropy weight to the uniform weight averaged over the batch.

Equivalently, the entropic adversary at temperature $\eta$ has the same effective support size as a hard CVaR adversary that selects the top $\lceil n/\Gamma_{\text{eff}} \rceil$ losses in each batch, although the two assign weights differently. Table 5 reports this mapping, evaluated on CIFAR-10H training batches of size $n=128$ using the trained Robust MoE checkpoint at each $\eta$ value.

*Table 5.* Empirical mapping between the entropic temperature $\eta$ and the effective density-ratio bound $\Gamma_{\text{eff}}$ on CIFAR-10H, batch size $n=128$. The effective fraction is the perplexity divided by the batch size; the equivalent hard-CVaR adversary averages the top $\lceil n/\Gamma_{\text{eff}} \rceil$ losses.

| $\eta$ | Perplexity | Effective fraction | $\Gamma_{\text{eff}}$ | Hard top-$\lceil n/\Gamma_{\text{eff}} \rceil$ equivalent |
|---|---|---|---|---|
| 0.5 | 122.4 | 95.6% | 1.0 | top 128 of 128 |
| 1.0 | 115.6 | 90.3% | 1.1 | top 117 of 128 |
| 1.5 | 106.6 | 83.3% | 1.2 | top 107 of 128 |
| 2.0 | 93.2 | 72.8% | 1.4 | top 92 of 128 |
| 2.5 | 83.4 | 65.1% | 1.5 | top 86 of 128 |
| 3.0 | 80.1 | 62.6% | 1.6 | top 80 of 128 |
| 4.0 | 67.4 | 52.6% | 1.9 | top 68 of 128 |
| 5.0 | 65.6 | 51.3% | 2.0 | top 64 of 128 |

Three observations follow from this mapping. First, the default $\eta = 2.0$ used throughout the experiments corresponds to $\Gamma_{\text{eff}} \approx 1.4$, i.e., a moderate tail emphasis equivalent to averaging over the top $\sim 73\%$ of losses in each batch—far from the extreme concentration regime $\Gamma \gg 10$ where smooth and hard tail risks diverge. Second, the mapping is monotone and smooth in $\eta$: doubling $\eta$ from 2.0 to 4.0 moves $\Gamma_{\text{eff}}$ only from 1.4 to 1.9. Third, the perplexity-derived $\Gamma_{\text{eff}}$ closely matches the hard top-$k$ batch fraction at the same nominal $\Gamma$, supporting the use of the tilted-softmax objective as a smooth surrogate for the hard CVaR objective in our operating regime.

# E. Additional figures/experiments

This appendix collects supporting figures for the experiments in §6. For each dataset we provide reliability diagrams for all methods, evaluated both on the full test set and on the dataset's hard subset (§D.5). We close with sensitivity ablations for the robustness parameter $\eta$ and the warmup horizon.

**CIFAR-10H**   The hard subset consists of CIFAR-10H test images with low human agreement, where the human soft labels themselves indicate genuine ambiguity. Figure E.2 shows representative examples where Vanilla MoE and MoCaE place near-unit confidence on a single class even though the human label distribution is spread over several classes; the robust methods correctly disperse mass over the plausible classes. Figure E.3 aggregates this pattern across the full test set: on easy images all methods produce sharp, high-confidence predictions; on hard images, only the robust methods produce a confidence distribution centered near the empirical accuracy, while the non-robust baselines remain peaked at high confidence.

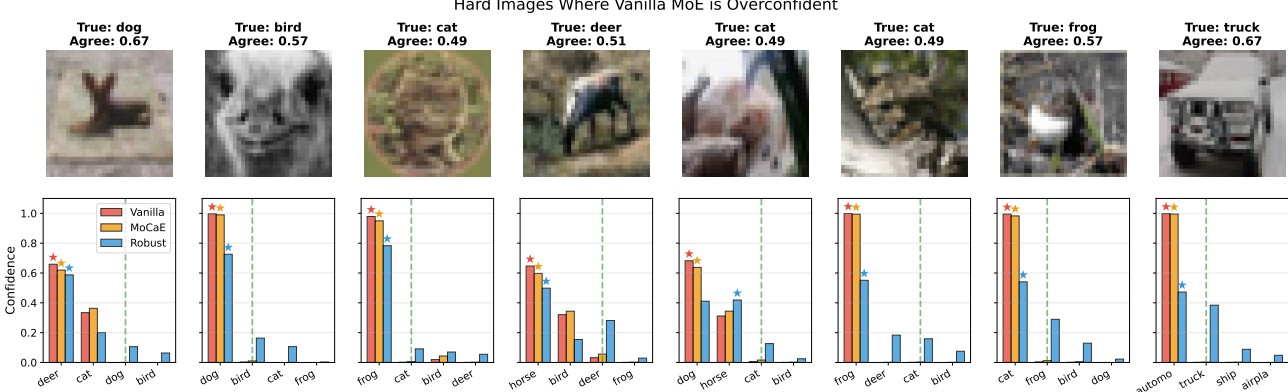

*Figure E.2.* Representative hard CIFAR-10H images on which the non-robust MoE baselines are confidently wrong. Each column shows one ambiguous image with the predicted confidence distribution over the top classes for each method. The robust methods reduce confidence on these images and place mass on multiple plausible classes, while the non-robust baselines remain peaked on a single class.

We report reliability diagrams for all methods in Figure E.4, evaluated separately on (a) the full test set and (b) the hard subset of low-agreement images. Panel (a) is an expanded view of Figure 1 that includes per-bin error bars, while panel (b) restricts the same diagnostic to the routing-stressed subset and is the most direct visual evidence for Takeaway 1 in §6: the non-robust baselines concentrate predictions in the highest-confidence bins despite low accuracy on those bins, while the robust methods spread mass over a wider confidence range that more closely tracks the diagonal.

**PACS**   Across all four target domains, the robust methods (Robust MoE and Robust Filtered) substantially improve calibration compared to the baselines, generally at a small cost to accuracy (Table 2). The reliability diagrams reinforce these findings and reveal a qualitative pattern beyond what ECE captures alone. On the harder domains (Art, Cartoon, Sketch), the baselines are all overconfident: most predictions fall in high-confidence bins (0.7-1.0), and since the low-confidence bins are nearly empty, they have large error bars that span much of the y-axis. In contrast, the robust methods distribute predictions more broadly across confidence bins, with tighter error bars and closer alignment to the diagonal.

Reliability diagrams for all methods across the four leave-one-domain-out target domains are shown in Figure E.5.

**CivilComments**   The CivilComments hard subset is the $\sim 41\%$ of test comments that mention at least one of eight identity subgroups (§D.5). Figure E.6 shows reliability diagrams on the full test set and on this hard subset. Both panels reproduce the qualitative pattern observed on CIFAR-10H: the non-robust MoE baselines concentrate predictions in the highest-confidence bin, where accuracy is below the bin-mean confidence, whereas the robust methods spread predictions across the confidence range and align more closely with the diagonal. The effect is most pronounced in panel (b), where Vanilla MoE and MoCaE remain confidently miscalibrated on identity-mentioning comments while Robust MoE and Robust Filtered are visibly closer to perfect calibration. This visual pattern matches the per-group accuracy and ECE breakdown in Figure 2 of §6.

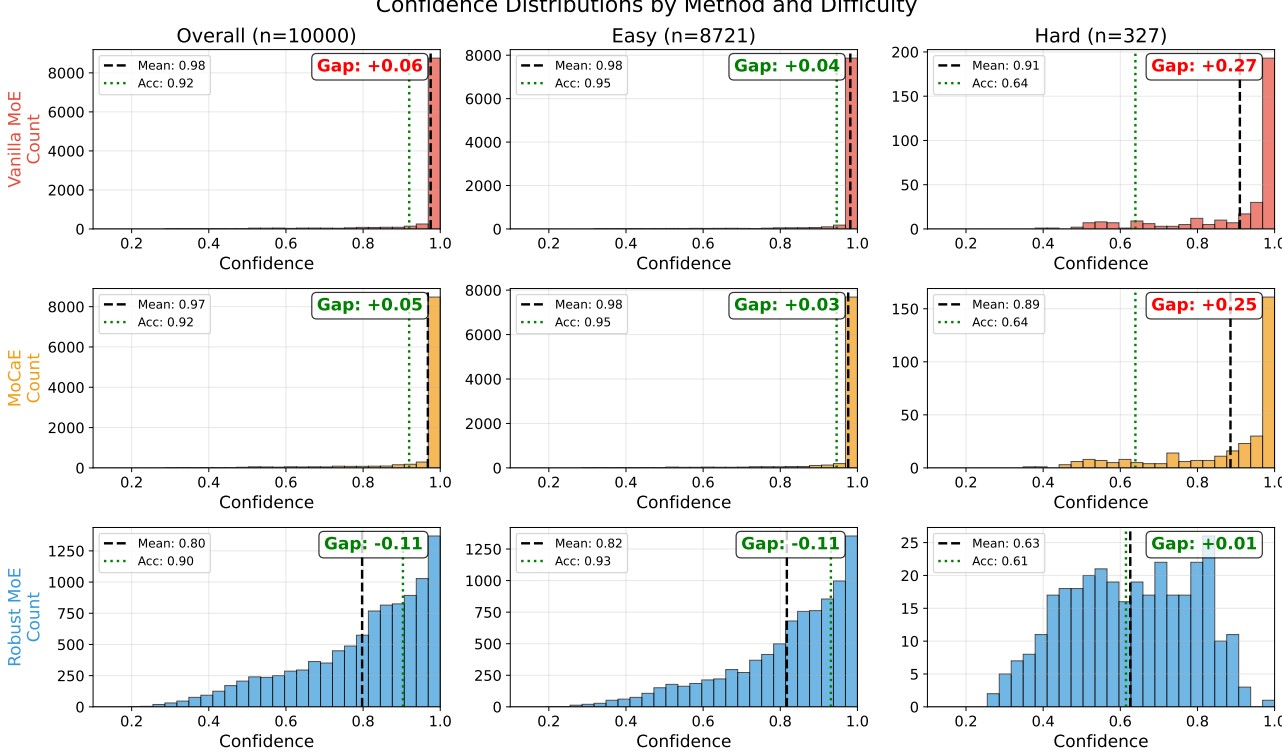

*Figure E.3.* Top-class confidence distributions for each method across CIFAR-10H difficulty levels. Each row corresponds to one method; columns split the test set into Overall, Easy (human agreement $> 0.9$), and Hard (human agreement $< 0.7$) subsets. Each panel reports the mean confidence (dashed) and accuracy (dotted) on the corresponding subset, and the gap between them (mean confidence minus accuracy). All methods are well calibrated on easy images. On the hard subset, the non-robust baselines remain concentrated at near-unit confidence despite accuracy near $0.6$, while the robust methods produce a wider confidence distribution whose mean tracks the empirical accuracy.

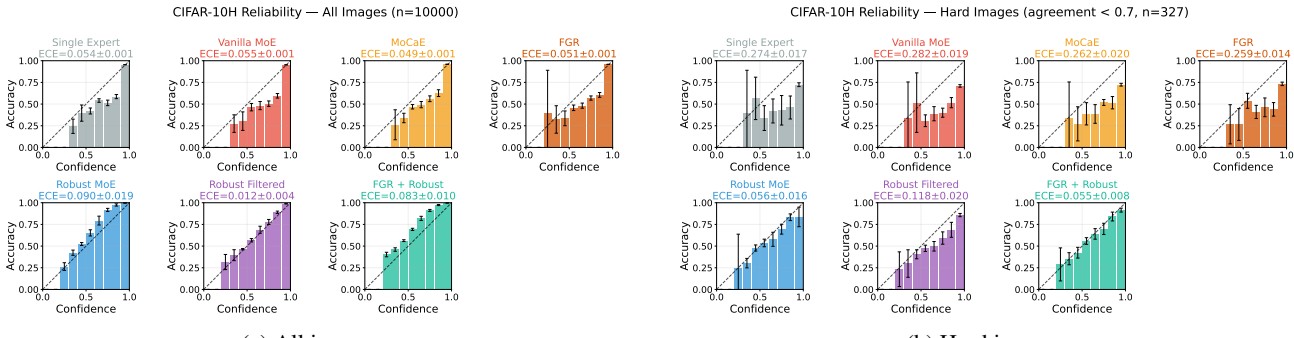

(a) All images.        (b) Hard images.

*Figure E.4.* Reliability diagrams for all methods on CIFAR-10H. (a) All test images, an expanded view of Figure 1 with per-bin error bars. (b) Hard subset (low-agreement images) only.

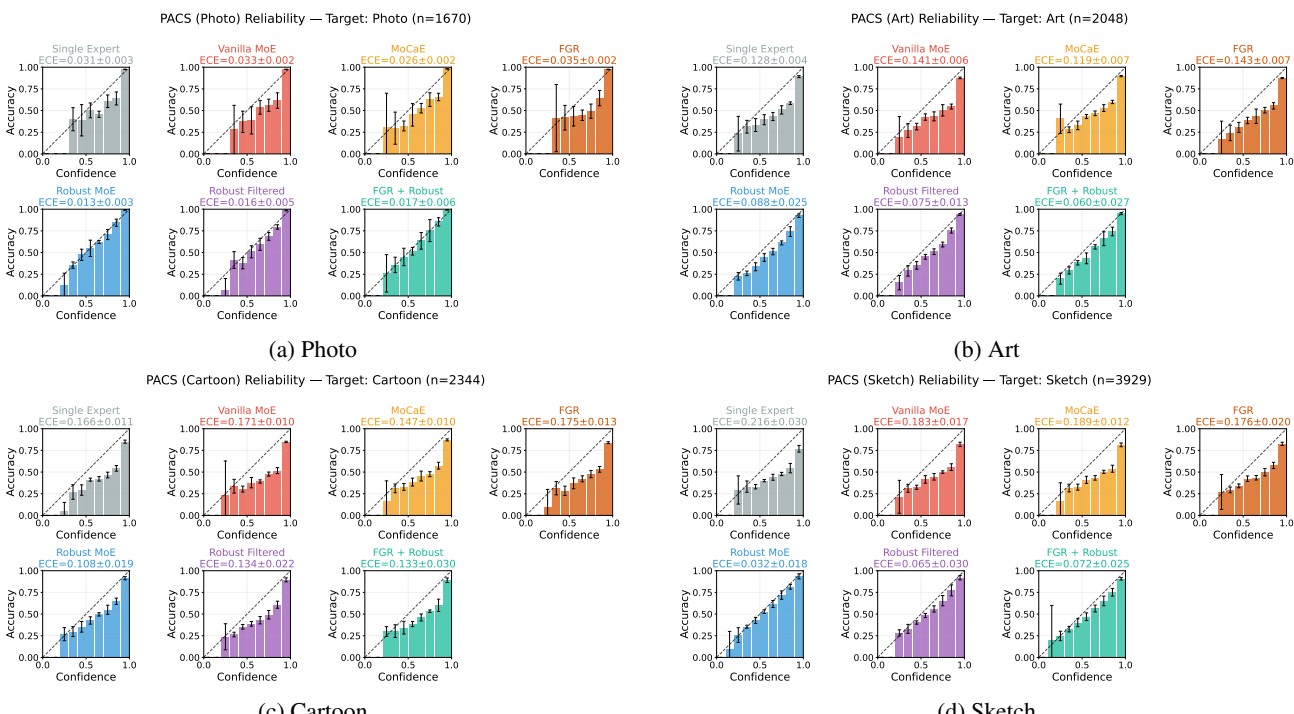

(a) Photo

(b) Art

(c) Cartoon

(d) Sketch

*Figure E.5.* Reliability diagrams for all methods on the four PACS target domains.

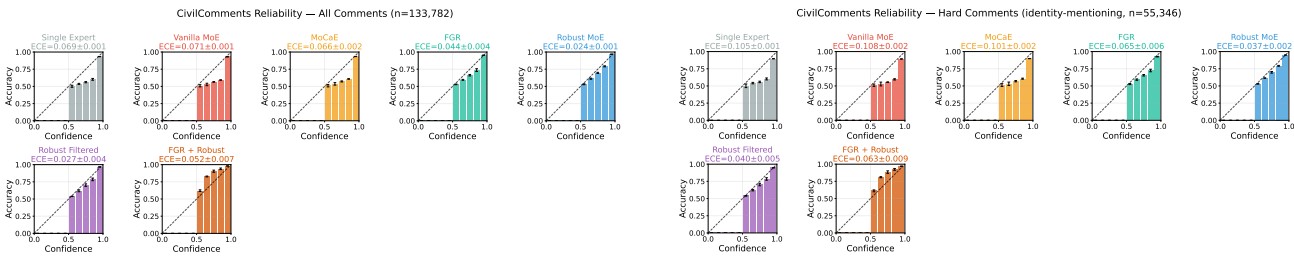

(a) All comments.

(b) Identity-mentioning comments.

*Figure E.6.* Reliability diagrams for all methods on CivilComments. (a) Full test set. (b) Hard subset of comments mentioning at least one identity subgroup.

**Sensitivity to $\eta$ and warmup.** The two hyperparameters specific to the proposed methods are the entropic temperature $\eta$, which controls the strength of the adversarial reweighting, and the warmup horizon $T_{\mathrm{warm}}$, which controls how long ERM precedes the robust objective. Figures E.7 and E.8 sweep each hyperparameter on CIFAR-10H, with all other settings fixed to the defaults reported in §D.4.

The default $\eta = 2.0$ used throughout the experiments sits near the Hard-ECE optimum and gives a small accuracy concession relative to ERM; both Hard ECE and accuracy degrade smoothly outside this neighborhood, with no sharp transitions. The warmup sweep shows that $T_{\mathrm{warm}} \in [15, 25]$ (out of 50 training epochs) gives the best joint calibration-accuracy tradeoff. Too little warmup applies the adversarial reweighting before features have stabilized and amplifies initialization noise; too much warmup leaves few epochs for the robust objective to take effect.

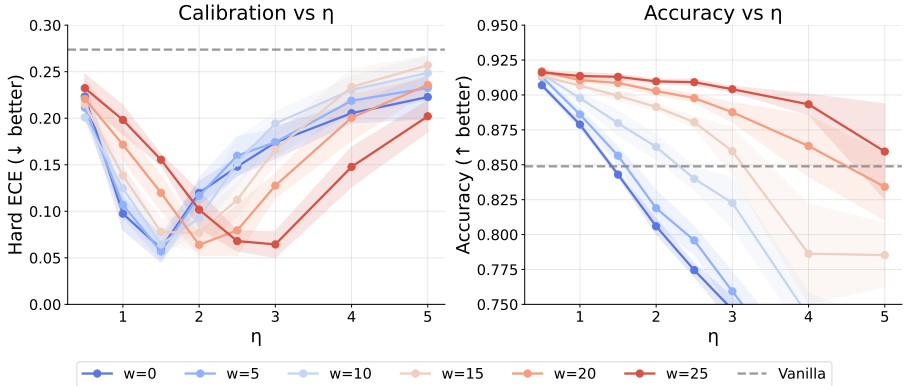

*Figure E.7.* Effect of the robustness parameter $\eta$ on calibration and accuracy on CIFAR-10H. Higher $\eta$ increases focus on high-loss samples during training, improving Hard ECE at the cost of overall accuracy.

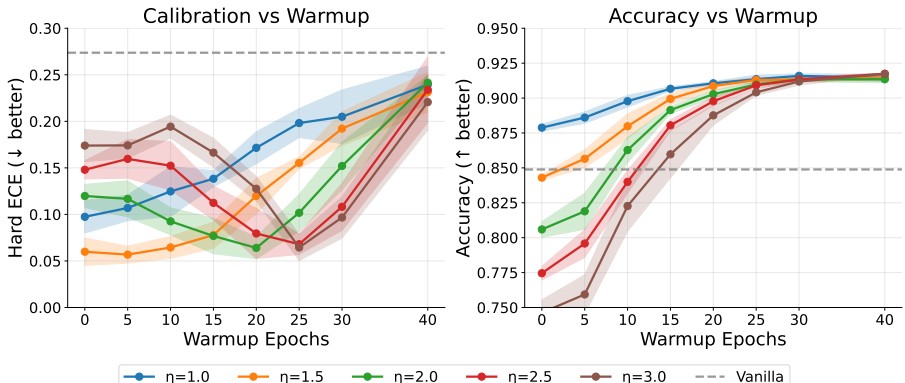

*Figure E.8.* Effect of warmup horizon $T_{\mathrm{warm}}$ on calibration and accuracy on CIFAR-10H. Warmup lets the model establish features before adversarial reweighting begins; $T_{\mathrm{warm}} \in [15, 25]$ epochs (out of 50) gives the best joint calibration-accuracy tradeoff.

