# OpenReview forum: "Toward Calibrated Mixture-of-Experts Under Distribution Shift"
_ICML.cc/2026/Conference — ICML 2026 regular_

### Official Review · Reviewer_5ZLA · 2026-03-08

**Soundness:** 1
**Presentation:** 3
**Significance:** 3
**Originality:** 3
**Overall Recommendation:** 4
**Confidence:** 4

**Summary:**

This paper studies calibration in mixture-of-experts (MoE) models under distribution shift. The authors show that even if each expert is perfectly calibrated on its own training distribution, the overall MoE model can become miscalibrated when the test distribution reweights the training distribution, i.e, distribution shift.

Specifically, if the router uses hard routing, calibration is preserved as long as the test data is not out-of-distribution. However, with soft routing, distribution shift can lead to miscalibration. The authors demonstrate through a small experiment that this issue arises not only in adversarial settings but also in common scenarios, such as when the difficulty distribution between training and test data differs slightly.

To address this issue, the authors propose training the router with a worst-case Brier loss to improve calibration. Experiments show that this approach significantly improves calibration in settings with mixed difficulty distributions.

**Compliance With Llm Reviewing Policy:**

Affirmed.

**Final Justification:**

I am satisfied with the revision plan proposed by the authors. Given that the revisions to the experimental section constitute a substantial undertaking, the authors are unable to complete them during the rebuttal phase.

upd: The authors added some experimental revisions, but these are still not enough to support the entire paper.

We are now faced with two options:

Accept the paper, albeit with the caveat that the entire experimental section remains un-peer-reviewed (with the authors undertaking the revisions during the camera-ready phase); or

Reject the paper due to the deficiencies in the experimental section and await the next ML conference.

Given that the theoretical contributions of this paper meet the acceptance threshold, I consider both options to be viable.

**Key Questions For Authors:**

See above

**Limitations:**

See above

**Strengths And Weaknesses:**

## Overall Evaluation

I like this paper very much. The key observation of the paper is important, and the method proposed is clear and implementable. However, the experimental section (both the setup and the results) are incomplete and crudely made. The paper also ends suddenly after the experiment section. In its current form, this paper does not meet the acceptance threshold.

Given I like the core part of this paper. Given that I like the core part of this paper, if the authors can substantially improve the experiments and present a more complete version of the work during rebuttal, I will increase my score to 5 (accept).

## Strengths

### 1. Important observation

Calibration of AI models is an important topic, and there is currently limited research on the MoE model calibration. I greatly appreciate the main observation of this paper, i.e., even if each expert is individually calibrated, the overall MoE model can still become miscalibrated. While the logic behind this observation is straightforward, it is somewhat unintuitive. Formalizing and rigorously proving this phenomenon is novel and valuable.

### 2. Simple and practical solution

The proposed solution, that training the router using a worst-case Brier loss, is simple and easy to implement. The experimental results suggest that this approach can effectively improve calibration.

### 3. Good writing (before the experiment)

The paper is generally well written and the flow is clear. However, some explanatory passages are unnecessarily long. For example: Line 109 (“Without calibration…”), Line 177 (“Example”), Line 267 (“Example”). These explanations might be replaced by a figure or written more concisely.

## Weaknesses

### 1. Experiment

The experimental section of this paper is its fatal weakness. In its current form, the experiment is incomplete and crudely made.

* **Overall experimental design.**
The paper evaluates only one task, one dataset, and one model, which makes the result not convincing enough. To show that distribution shift can affect calibration in MoE models, it would be helpful to include additional tasks, datasets, or architectures to strengthen the claim.

* **Dataset choice.**
The experiments use CIFAR-10H as the test set. However, to construct the distribution shift, only the hardest 3% samples are used as the test set. This means the “HARD” test set contains only 300 samples. It is well known that ECE cannot be reliably estimated with such a small number of samples. Despite this, there is a minor issue that CIFAR-10H may have different distribution from CIFAR-10 (training set).

* **Artificial distribution shift.**
The distribution shift is constructed by manually selecting the hardest 3% samples from CIFAR-10H. This is quite artificial and does not resemble a natural distribution shift.

* **Choice of hyperparameter w.**
The main paper only reports results for w=20, while put the results of other choices in the appendix. I do not think this should be in the appendix. In addition, the authors should discuss how to select w in practice in the main text.

* **Experiment does not demonstrate the advantage of MoE.**
The experiments use 4xResNet-18 as experts. This is a very small and toy MoE setting. Moreover, the results show that the MoE model performs worse than a single ResNet-18, which is confusing and suspicious. It would be better to evaluate the method in a setting where MoE is more appropriate.

* **Presentation of the results.**
Figure 1 occupies a large amount of space but appears to be only an example. Only one subfigure is sufficient enough. Figure 2 and Table 1 contain the main results. It would be helpful to plot both proper loss and calibration error across the five splits in a figure similar to Figure 2, and to include the temperature scaling (TS) baseline in that figure.

### 2. Assumption on distribution shift

The distribution shift model assumes that an adversary can upweight samples up to Γ times. The proposed worst-case optimization depends on this assumption and the Γ. Since Γ determines the parameter w, it would be helpful to discuss how to estimate an appropriate value of w in low sample complexity given train and test distribution.

### 3. Missing explanations

Several aspects of the paper need clarification.

* Why is Brier loss used for the worst-case optimization of the soft router instead of other proper scoring rules?

* In the experiments, only Hard ECE is reported. Why is the ECE over the full test set not reported?

* In the experiments, what architecture is used for the router, and how the worst-case Brier loss optimization is implemented in practice.

* In the experiments, the train and val come from the same distribution, while the test is shifted. Why is the val not drawn from the same distribution as the test? This design may significantly affect the performance of temperature scaling.

### 4. Lack of discussion

The paper ends very abruptly after the experimental section. More discussion would be helpful, and I strongly suggest providing a clear takeaway message for readers.

Some potentially important discussion points include:

* What is the cost of Robust MoE? For example, how much performance is lost on in-distribution data (in terms of both loss and calibration)?

* In practical MoE systems, top-2 routing is widely used. Are there better solutions for this type of router?

* The paper currently studies routers only at the outermost level of the model. In many real systems (e.g., LLM MoE layers), routers appear inside the model. It would be interesting to discuss whether calibration improvements in internal MoE layers translate to improvements in the final model output, and how the proposed method could extend to this setting.

### 5. Missing impact statement

The paper appears to be missing the required impact statement.

### 6. Missing related work

The authors should put the related work in the main text rather than in the appendix.

--------

**Post Rebuttal**: Raised my score to 4.

---

> ### Author Rebuttal · Authors · 2026-03-31
>
> Thank you for the thoughtful review. We really appreciate that you find the main idea compelling and the proposed objective simple and implementable, and we are encouraged by your comment that a more complete empirical evaluation could significantly improve your assessment. While we think this will be best handled in the next revision, we do want to address your questions here and clarify how we plan to strengthen the paper.
>
> **On the experimental section**
>
> Your suggestions are very helpful here. Our goal in the current version was to isolate the routing-calibration mechanism in a controlled setting, which is why we used a relatively simple MoE and the CIFAR-10H construction. That gave us a clean environment in which the effect is visible, but it also made the paper read more narrowly than the main idea warrants. In revision, one of our main priorities will be to broaden and reorganize the experimental story so that the MoE calibration issue we identify is supported more fully across different settings.
>
> **On Brier loss and the robustness parameter**
>
> We want to clarify that the MoE calibration issue is not specific to the Brier score. That issue arises because the mixture weights change under shift, and occurs at the level of the predicted probabilities themselves. For that reason, the same issue would still show up if calibration were evaluated with another strictly proper scoring rule. We use the Brier score because, compared to other proper scoring rules, it leads to a cleaner and more stable objective in this setting. In particular, under adversarial reweighting, log loss can be dominated by a very small number of extreme-confidence mistakes, whereas Brier gives a more stable tail-risk objective and a more manageable optimization problem.
>
> The robustness parameter plays a similar role. We do not view it as inducing a uniform reduction in confidence; rather, it moves the model along an accuracy/calibration frontier by redistributing confidence away from slices where the soft-routed aggregate is brittle under shift. If set too aggressively, it can indeed trade some average-case sharpness for performance on those vulnerable regions. In practice, we find the movement in average performance to be relatively modest; we will characterize this more thoroughly in revision.
>
> **On experimental details**
>
> Thank you for highlighting the places where the draft is currently too terse; we hope to add fuller discussion in the revision.
> - The router networks are two-layer MLPs, ending in a softmax layer.
> - For the robust objective, we compute per-example Brier losses, select the top-loss $\lceil n/\Gamma \rceil$ within each batch, and backpropagate through their mean.
> - Our held-out validation split is drawn from the training distribution. We agree that temperature scaling would likely perform better if validation had the same distribution as test, but that would also assume some access to test-time information.

---

> > ### Author Rebuttal · Reviewer_5ZLA · 2026-03-31
> >
> > I am satisfied with the revision plan proposed by the authors. Given that the revisions to the experimental section constitute a substantial undertaking, the authors are unable to complete them during the rebuttal phase. We are now faced with two options:
> >
> > 1. Accept the paper, albeit with the caveat that the entire experimental section remains un-peer-reviewed (with the authors undertaking the revisions during the camera-ready phase); or
> >
> > 2. Reject the paper due to the deficiencies in the experimental section and await the next ML conference.
> >
> > Given that the theoretical contributions of this paper meet the acceptance threshold, I consider both options to be viable.
> >
> > Furthermore, I hope the author will respond to the "Lack of discussion" section I raised.
> >
> > I am raising my score to 4 and strongly urge the authors to proceed with the revisions in accordance with the provided revision plan.

---

> > > ### Author Response · Authors · 2026-04-06
> > >
> > > Thank you for your follow-up. We really appreciate your positive assessment of our work, and we thank you for your willingness to engage with our revision plan.
> > >
> > > We do take your feedback on the limited experiments seriously, and we have been working to substantially expand our experiments since our initial rebuttal. We would like to share our current progress at https://dblindanon.github.io
> > >
> > > For these experiments, we also introduce a new variant of our proposed method, *Robust Filtered*, which restricts the robust tail emphasis to examples where routing is actually relevant---specifically, where the mixture prediction underperforms its best expert, or where experts meaningfully disagree. This makes the robustness objective more targeted and, in practice, largely resolves the accuracy/calibration tradeoff present in the original method *Robust MoE*. The website includes an overview of the method (R.1).
> > >
> > > We summarize the key findings below:
> > >
> > > - **CIFAR-10H (expanded).** We now report overall ECE alongside Hard ECE. We show that both robust methods (Robust MoE and Robust Filtered) achieve large reductions in Hard ECE, while Robust Filtered additionally achieves the best overall ECE. This shows that the improvement on the hard samples does not necessarily need to come at a cost to in-distribution performance.
> > >
> > > - **PACS: domain shift (new).** We evaluate on the PACS domain generalization benchmark using the standard leave-one-domain-out protocol. Again, the robust methods substantially improve calibration across all four target domains, with minimal cost to accuracy.
> > >
> > > - **CivilComments: text with subpopulation shift (new).** We extend the framework to a transformer-based architecture on the WILDS CivilComments dataset. We show the robust methods substantially improve calibration with no accuracy cost, and this improvement holds across all 8 demographic groups in the dataset.
> > >
> > > We hope that these experiments help to address some of your core concerns: they span multiple datasets (CIFAR-10H, PACS, CivilComments), multiple tasks (image classification, domain generalization, text toxicity detection), multiple models (ResNet-18, pretrained and otherwise; DistilBERT), and both artificial and natural distribution shifts. We also now report overall ECE alongside subgroup metrics throughout.
> > >
> > > ---
> > > **Additional discussion**
> > >
> > > We fully agree that we should have a discussion section to better explain the utility/limitations of our approach. We plan to add a discussion section to the revision; in the meantime, we would like to respond to the points you raised directly.
> > >
> > > *What is the cost of Robust MoE on in-distribution data?*
> > >
> > > The cost depends on the data. The robust objective upweights the highest-loss examples, so when the hard subpopulation is small (as in CIFAR-10H, ~3% of test data), optimization concentrates on a narrow region and Robust MoE trades some in-distribution ECE for Hard ECE improvement. Robust Filtered addresses this by restricting the robust emphasis to routing-relevant examples only.
> > >
> > > When the hard subpopulation is larger or the shift is more natural, the tradeoff diminishes: the high-loss examples are no longer rare outliers but a representative part of the data, so improving calibration on them generalizes more broadly. In CivilComments and PACS, the robust methods improve both overall and subgroup calibration with little to no accuracy cost.
> > >
> > > *Are there better solutions for top-2 routing?*
> > >
> > > Our robust objective applies directly to top-2 routing, since top-2 is still a soft mixture with learned weights subject to routing posterior shift. But top-2 is also a natural fit for a more targeted approach: with only two active experts per input, the routing-relevant region---where experts disagree or the mixture underperforms the better expert---can be identified cleanly. In dense soft routing, this filtering is useful but fuzzier since all experts contribute; in top-2, the structure makes it a direct diagnostic.
> > >
> > > *Do calibration improvements in internal MoE layers translate to improvements in the final model output?*
> > >
> > > Our current analysis focuses on MoE at the prediction level, where the connection between routing and output calibration is direct. In deep MoE architectures where routing occurs at internal layers, this relationship is less clear. That said, we believe internal routing fragility still matters: when experts at an internal layer would produce meaningfully different hidden states, instability in routing weights under shift changes the representation that downstream layers operate on. This becomes less about mixture-level calibration and more about stability of the routed representation under shift. Testing this is nontrivial---isolating internal routing instability from other sources of instability requires careful experimental design---but it is a practically important question.
> > >
> > > ---
> > > Thank you again for your constructive and generous engagement with our work.

---

### Official Review · Reviewer_7bYD · 2026-03-13

**Soundness:** 3
**Presentation:** 3
**Significance:** 2
**Originality:** 3
**Overall Recommendation:** 3
**Confidence:** 3

**Summary:**

This paper studies calibration in Mixture-of-Experts (MoE) models under distribution shift. The authors argue that for hard routing, if each expert is calibrated on its own region, the overall model remains calibrated under a broad class of reweighted distributions. However, as for soft routing, the pooled prediction can become miscalibrated even when individual experts are calibrated. The reason is a mismatch between the routing posteriors at training and test time. To address this issue, they propose a training objective based on a distributionally robust proper scoring rule. It emphasizes the most vulnerable high-loss regions using a bounded density ratio formulation. This improves calibration for soft-routed MoEs under shift.  Based on these motivations, they validate their method on CIFAR10H, where human labeling consistency is used to construct latent difficulty. They compare a single expert, vanilla MoE, MoCaE, and their proposed robust MoE. Results show that their method substantially reduces calibration error on the “hard/ambiguous” subset. This improvement comes with some trade-off in overall accuracy or overall calibration.

**Compliance With Llm Reviewing Policy:**

Affirmed.

**Final Justification:**

This paper studies calibration in Mixture-of-Experts models under distribution shift. To address this problem, the authors propose several routing strategies and validate their method on CIFAR-10H. However, the proposed approach is intended for a relatively general setting, it is necessary to be evaluated on more datasets seems . Otherwise, the method may appear overly tailored to CIFAR-10H. In addition, the paper should provide more analysis of the key threshold hyperparameters used in the experiments to rule out gains. The overall presentation also needs improvement, especially in the quality of some figures. Based on these considerations, I assign the paper a score of 3.

**Key Questions For Authors:**

- Beyond CIFAR10H, can the authors add more shift benchmarks?
- Can the authors include stronger and more relevant baselines that are designed specifically for calibration under distribution shift or for improving MoE routing?
- More ablation experiments for the routing posterior shift is important.

**Limitations:**

yes

**Strengths And Weaknesses:**

**Strengths:**

- This pipeline is coherent: it moves from an analysis of hard routing, to the failure mode under soft routing, and then to a distributionally robust Brier objective that targets the most vulnerable regions.
- In CIFAR10H experiments, Robust MoE reduces Hard ECE from around 0.27 to 0.064. This gain persists even after temperature scaling.

**Weaknesses:**

- The whole experiment is conducted almost entirely on CIFAR10H. It is too single. Therefore, I believe more extensive experiments are needed to validate this general MoE calibration method.
- It is a little subjective for the "routing posterior shift" setting (agreement threshold). The chosen threshold inevitably affects the results. To address this, the authors should introduce more shift variants in experiments and add these results into the ablation section.
- The baseline comparisons are insufficiently strong. This makes the conclusions less robust. In this paper, the authors only compare against a single expert, vanilla MoE, and a modified version of MoCaE. However, MoCaE originates from object detection and it is not a direct baseline designed for the calibration-under-distribution-shift problem addressed here. At the same time, recent work on robust calibration under distribution shift, subpopulation calibration, and improvements to MoE routing mechanisms offers more relevant baselines for comparison [1,2,3].
- The method improves tail / hard-slice calibration, but is not better overall.

[1] Zhang, Yilin, et al. "Gradient Rectification for Robust Calibration under Distribution Shift." *arXiv preprint arXiv:2508.19830* (2025).

[2] Roschewitz, Mélanie, et al. "Where are we with calibration under dataset shift in image classification?." *arXiv preprint arXiv:2507.07780* (2025).

[3] Wang, Ziteng, Jun Zhu, and Jianfei Chen. "Remoe: Fully differentiable mixture-of-experts with relu routing." *arXiv preprint arXiv:2412.14711* (2024).

---

> ### Author Rebuttal · Authors · 2026-03-31
>
> Thank you for the thoughtful review. We are glad that you find our story coherent, and that you see the proposed solution as a natural response to the problem we identify.
>
> **On the experimental scope**
>
> We agree with your point that the experimental section is the place where the current version is least complete. Our main observation is that soft-routed MoEs can become miscalibrated under distribution shift even when the experts themselves are calibrated, and the primary aim in our experimental section was to isolate the mechanism as cleanly as possible. CIFAR-10H was useful for this because it gives a controlled way to study ambiguity-driven shift through human agreement annotations. At the same time, you are right that this setting is not, by itself, broad enough to support the wider empirical framing of the paper. In revision, we plan to strengthen with additional datasets, additional shift constructions, and settings where MoE is a more natural architectural fit.
>
> **On the "routing posterior shift" construction**
>
> We understand the concern that the current CIFAR-10H experiment uses an agreement threshold, and may therefore appear tied to a particular choice of cutoff. We want to clarify that our intended claim is not about one particular threshold, and that the threshold is just a convenient way to create a broader class of reweighting shifts that place more mass on higher-ambiguity examples. What matters for the experiment is just the relative increase in weight on regions where the soft-routed aggregate is most vulnerable, not the exact boundary used to define those regions. As you suggest, we will demonstrate this by adding ablations over the threshold choice, and by including shift variants that vary the ambiguity mix more smoothly.
>
> **On tail calibration versus overall performance**
>
> The proposed method does focus on improving performance on the hard slice rather than uniformly / in aggregate. This is because the failure mode we study---the fragility of soft routing---is concentrated in the overlap regions where the experts disagree, and the robust objective is specifically designed to train against that vulnerability. Empirically, we see that this leads to large gains on the hard subset, where calibration improves substantially, but with less change in the overall performance. We thank you for pointing this out, and will clarify this balance in revision.

---

> > ### Author Rebuttal · Reviewer_7bYD · 2026-04-01
> >
> > I believe the authors should strengthen the paper by conducting the **necessary experiments** rather than relying solely on descriptive statements.
> >
> > Specifically:
> >
> > **(1)** Since the paper targets a relatively **general algorithm and framework**, it would be valuable to include **additional benchmarks** to further validate its effectiveness and generality.
> >
> > **(2)** The sensitivity and robustness of the threshold setting should be examined in the current CIFAR-10H experiments, as this factor has an important impact on the stability of the proposed method.
> >
> > **(3)** In addition, there already exist several relevant **baseline methods** for **MoE with expert routing calibration**. It is necessary to compare the proposed approach with these methods and provide a more detailed discussion of their relative advantages and limitations.

---

> > > ### Author Response · Authors · 2026-04-08
> > >
> > > Thank you for your follow-up. We understand your concerns and we agree that, for a method intended as a general algorithmic framework, full empirical validation is important. Following your feedback, we used the discussion period to implement and run additional experiments, with new results available at https://dblindanon.github.io/
> > >
> > > We hope these experiments help address your main concerns. They span multiple datasets (CIFAR-10H, PACS, CivilComments), multiple tasks (image classification, domain generalization, text toxicity detection), multiple models (ResNet-18, pretrained and otherwise; DistilBERT), and include the three additional baselines you suggested. We also added a sensitivity analysis for the threshold hyperparameter in the CIFAR-10H setting, following your suggestion.
> > >
> > > We wanted to address your concerns as fully as possible, and we hope you have a chance to look over the new results. Our sincere thanks for your time and engagement.

---

### Official Review · Reviewer_pWkB · 2026-03-13

**Soundness:** 3
**Presentation:** 3
**Significance:** 2
**Originality:** 3
**Overall Recommendation:** 3
**Confidence:** 5

**Summary:**

This paper studies the calibration properties of Mixture-of-Experts (MoE) models under distribution shift, focusing on how routing mechanisms interact with expert level calibration. The authors show that hard routed MoEs preserve calibration under a broad class of distribution shifts, while soft routed MoEs can become miscalibrated even when all experts are individually calibrated. This failure is attributed to shifts in routing induced mixture weights. To mitigate this issue, the paper proposes optimizing a distributionally robust proper scoring objective that regularizes the aggregate predictor against adversarial reweightings of the training distribution.

**Compliance With Llm Reviewing Policy:**

Affirmed.

**Final Justification:**

The reviewers addressed my concerns and I will be maintaining my score.

**Key Questions For Authors:**

1. How does routing posterior shift fundamentally differ from classical mixture weight shift analyses in mixture models or Bayesian model averaging? What is new beyond MoE-specific framing?
2. Can the authors provide an impossibility result or lower bound showing that no fixed soft-routed MoE can remain calibrated under arbitrary latent mixture reweighting, even with perfectly calibrated experts?
3. What minimal additional assumptions on the router would restore calibration guarantees for soft routing?
4. How sensitive are the results to the choice of the reweighting budget, and how should it be selected in practice?
5. Do the authors expect the same phenomena and benefits to hold in large-scale MoEs (e.g., language models with token-level routing)? What are the main obstacles to demonstrating this empirically?

**Limitations:**

Impact statement is missing. Given the nature of the work dealing with calibration of mixture models with the intent of robustifying, I think it is important for the authors to discuss the limitations of the proposed approach along with societal impacts of improperly calibrated mixture models.

**Strengths And Weaknesses:**

Strengths:

1. The distinction between hard and soft routing is well discussed, and the paper provides an intuitive explanation for why hard routing induces an information bottleneck that protects calibration under reweighting shifts, whereas soft routing does not.

2. The characterization of calibration preserving shifts via sufficient statistics of the model output is clearly presented. The notion of routing posterior shift is internally consistent and well motivated.
3. Framing calibration robustness through a distributionally robust proper scoring rule is reasonable, and the CVaR interpretation of the objective is technically correct. The connection to multiaccuracy provides additional theoretical context.


Weaknesses:


1. The core insight that soft mixtures are fragile to shifts in mixture proportions even when components are well calibrated is well known in the broader literature on mixture models. Further, the theoretical contributions are largely descriptive, explaining why calibration fails under soft routing rather than establishing theoretical results such as impossibility results etc,.

2. The CVaR style distributionally robust objective is a standard robustness technique used in many contexts such as robustness, fairness, tail risk, calibration. The paper does not fully justify why this approach is uniquely suited to MoEs.
3. Experiments are limited to CIFAR 10H with small MoEs and linear experts. There is no evaluation on larger scale or more realistic MoE settings (e.g., language or vision transformers), which limits practical impact.
4. While the Brier score is convenient, many arguments appear to rely only on strict propriety rather than quadratic structure. It is unclear whether the conclusions depend meaningfully on using Brier versus log loss or other proper scoring rules.

---

> ### Author Rebuttal · Authors · 2026-03-31
>
> Thank you for the thoughtful review. We are glad that you appreciated how we characterized calibration-preserving shifts using sufficient statistics of the model output, and that you find our notion of routing posterior shift to be well-motivated.
>
> **Novelty relative to broader literature on mixture models**
>
> We agree that the core insight is intuitive, especially to researchers who have worked closely on mixture models. At the same time, to the best of our knowledge, this calibration question has not been directly isolated for learned MoEs in the form we study here:
> (i) the explicit contrast between hard and soft routing, where hard routing preserves calibration under region reweighting but soft routing can fail even when the experts themselves are calibrated
> (ii) the identification of the relevant calibration-preserving statistic in the two cases
> (iii) the connection from that failure mode to a practical robust training objective
> We are happy to make this positioning more explicit in revision.
>
> **Impossibility result**
>
> Consider a binary soft-routed MoE with fixed experts $f_1,f_2:\mathcal X\to[0,1]$ and fixed router $g:\mathcal X\to[0,1]$, with prediction $m(x)=g(x)f_1(x)+(1-g(x))f_2(x)$.
> Assume the experts are non-identical, i.e. $\mathbb P\big(f_1(X)\neq f_2(X)\big)>0$.
>
> Under a shifted distribution $P'$, keep the marginal of $X$ unchanged, keep the expert-conditionals fixed so that $\mathbb E_{P'}[Y\mid X=x,Z=k]=f_k(x)$, but change the latent posterior to some $g'(x)=P'(Z=1\mid X=x)$. Then the true conditional label probability becomes $\eta'(x)=g'(x)f_1(x)+(1-g'(x))f_2(x)$, so the deployed MoE has pointwise error
> $$
> \eta'(x)-m(x)=(g'(x)-g(x))(f_1(x)-f_2(x)).
> $$
>
> Now define two extreme reweightings:
> $$
> g_+(x)=\mathbf 1[f_1(x)>f_2(x)],\qquad
> g_-(x)=\mathbf 1[f_1(x)<f_2(x)].
> $$
> For $d(x):=f_1(x)-f_2(x)$, these give
> $$
> (g_+(x)-g(x))\,d(x)\ge 0,\qquad
> (g_-(x)-g(x))\,d(x)\le 0
> \quad\text{for all }x.
> $$
> Moreover, because $\mathbb P(d(X)\neq 0)>0$, at least one of these inequalities is strict on a set of positive measure. Hence for at least one choice $g'\in\{g_+,g_-\}$, $\mathbb E_{P'}[\eta'(X)-m(X)] \neq 0$. But if $m$ were calibrated under $P'$, we would have $\mathbb E_{P'}[Y\mid m(X)] = m(X)$ a.s., which implies $\mathbb E_{P'}[\eta'(X)-m(X)] = 0$, a contradiction. Therefore, for any nontrivial soft-routed MoE with non-identical experts, there exists a latent mixture reweighting under which the frozen MoE is miscalibrated, even though the experts remain perfectly calibrated.
>
> **CVaR objective**
>
> As you observe, CVaR-style worst-case reweighting is a standard robustness method that is not unique to MoEs, but it as a natural fit for the kind of shift we consider. Our shift model is a bounded population reweighting, and the fragility of soft-routing appears most strongly in the rare routing-overlap regions that average-risk training can ignore. A worst-case proper-scoring objective directly targets those rare high-loss slices without requiring us to identify them in advance.
>
> **Brier score**
>
> The main issue, that soft-routed MoEs can become miscalibrated under reweighting shift, does not depend specifically on the Brier score. That issue arises because the mixture weights change under shift, and occurs at the level of the predicted probabilities themselves. For that reason, the same issue would still show up if calibration were evaluated with another strictly proper scoring rule.
>
> We use Brier because it is more stable for the robust training objective; log loss is less convenient here because under adversarial reweighting, it can be dominated by a small number of extreme-confidence mistakes.
>
> **Sensitivity to the reweighting budget**
>
> In our experiments, we find that increasing the budget $\Gamma$ improves calibration on the hard shifted subset, but making $\Gamma$ too aggressive hurts overall performance. This is a gradual effect, e.g. with smaller budget of $\Gamma=5$ we have a Hard ECE of $0.057$, versus $0.064$ for $\Gamma=20$, while Hard Accuracy drops from $0.617$ to $0.543$. Empirically, this suggests that the method is not highly fragile to the exact value of $\Gamma$, but the choice of $\Gamma$ does move the operating point along the accuracy/calibration frontier. In practice, we select $\Gamma$ by validation over a small grid of candidate values.
>
> **Large-scale MoEs**
>
> We do expect the same issue to persist in larger MoEs, including token-routed language models, whenever prediction depends on effective mixture weights that can shift between train and test. The main empirical obstacle is isolating this issue for demonstration: because routing is internal and token-level, calibration at the final output is confounded by many other factors (sequence context, decoding, capacity effects, etc.) In addition, the vulnerable shifted slices are much harder to define. Thus, the main obstacle seems to be demonstrating the mechanism cleanly at scale, rather than a reason to think the mechanism disappears.

---

> > ### Author Rebuttal · Reviewer_pWkB · 2026-04-03
> >
> > I thank the authors for their responses, I don't have any further questions!

---

> > > ### Author Response · Authors · 2026-04-06
> > >
> > > Thank you for your follow up!
> > >
> > > Earlier, you had brought up a concern that our experiments were too limited to properly understand the practical impact of our method. We have been working on significantly expanding our experiments since then. Please feel free to check on our current progress at: https://dblindanon.github.io/
> > >
> > > ---
> > > We hope this helps to answer some of your previous concerns. In particular:
> > >
> > > *"The CVaR style distributionally robust objective is a standard robustness technique used in many contexts such as robustness, fairness, tail risk, calibration. The paper does not fully justify why this approach is uniquely suited to MoEs."*
> > >
> > > For these experiments, we introduce a new variant of our proposed method, Robust Filtered, which restricts the robust tail emphasis to examples where routing is actually relevant---specifically, where the mixture prediction underperforms its best expert, or where experts meaningfully disagree. The website includes an overview of this method in R.1. We maintain that our proposed objective, Robust MoE, targets an MoE-specific routing failure; however, we believe Robust Filtered does an even better job of distinguishing when a failure is MoE-related or not, and is therefore more uniquely suited to MoEs.
> > >
> > > *"Experiments are limited to CIFAR 10H with small MoEs and linear experts. There is no evaluation on larger scale or more realistic MoE settings (e.g., language or vision transformers), which limits practical impact."*
> > >
> > > Our new experiments include results on the PACS image dataset and the CivilComments WILDS text dataset. The distribution shifts in these datasets are more realistic, as PACS is a classic example of domain/style shift, and CivilComments is a natural example of subpopulation shift. We use the language model DistilBERT for the CivilComments dataset; we are working with limited compute, but we particularly wanted to incorporate at least one larger model following your suggestion. In our experiments, we see that both robust methods (Robust MoE and Robust Filtered) significantly improve calibration on all datasets for only a small drop in accuracy---and in the case of CivilComments, at no cost to accuracy, achieving Pareto improvement.
> > >
> > > ---
> > > We kept your feedback in mind as we made these experiments, and we hope you find this helps to demonstrate that our method works across a larger range of practical settings.

---

### Official Review · Reviewer_zHYU · 2026-03-22

**Soundness:** 3
**Presentation:** 3
**Significance:** 3
**Originality:** 3
**Overall Recommendation:** 4
**Confidence:** 4

**Summary:**

## Summary

This paper studies calibration in mixture-of-experts (MoE) models under distribution shift. The manuscript considers the key question of whether calibrating individual experts is enough to ensure calibration of the overall model when the data distribution changes. The work appears to examine a major question, especially given how widely MoE architectures are used today.

The main claim is a clear contrast between hard and soft routing. For hard routing, expert-level calibration is sufficient to preserve calibration under a class of distribution shifts (essentially reweighting of routing regions). For soft routing, however, even perfectly calibrated experts can lead to a miscalibrated overall predictor when the routing weights no longer match the test-time distribution. The paper attributes this to a mismatch in routing-induced mixture weights under shift.

To address this, the authors propose a distributionally robust objective based on proper scoring rules (Brier score), implemented via a CVaR-style top-loss reweighting. Empirically, they show that this significantly improves calibration on ambiguous examples in CIFAR-10H, where standard MoE and MoCaE baselines are overconfident.

**Compliance With Llm Reviewing Policy:**

Affirmed.

**Key Questions For Authors:**

## Questions for Authors

1. How essential is the interpretation of the router as approximating ($P(Z \mid X)$)? Would the failure mode still hold under a purely functional view of routing (i.e., without assuming a latent variable model)?

2. How should one choose the robustness parameter ($Γ$ or tail fraction) in practice? Is there any guidance beyond tuning?

3. Does the method systematically reduce prediction sharpness or confidence? If so, how should this tradeoff be controlled?

4. Have you tried this on larger MoE systems (e.g., with top-k routing or language models)? Do the same effects appear?

5. To what extent does the approach help under shifts beyond reweighting (e.g., label shift or changes in ($P(Y \mid X))$)?

**Limitations:**

- Assumes shifts are bounded reweightings, which may not reflect real-world changes
- Relies on a latent mixture interpretation that may not hold in practice
- Does not address routing instability from optimization dynamics
- Limited empirical validation beyond a single dataset
- Potential tradeoff between calibration and predictive sharpness not fully explored

**Strengths And Weaknesses:**

## Strengths

The paper is built around a simple but quite compelling idea: soft routing introduces a new source of calibration failure that is not present in hard routing. This distinction is clearly explained and, at least at a high level, convincing.

I also think the framing is useful. The paper isolates a failure mode that is not due to miscalibrated experts or standard covariate shift, but rather due to the interaction between routing and distribution shift. This is a meaningful contribution, especially given how often MoE is used in large systems.

On the method side, the proposed fix is straightforward and practical. The DRO objective reduces to a top-loss reweighting scheme that is easy to implement and does not require modeling the test-time shift explicitly.

Empirically, the results are consistent with the claims. In particular, it is interesting (and somewhat surprising) that calibrating experts alone (MoCaE) does not fix the issue, while the robust objective does.

---

## Weaknesses

The main issue I have is that the theoretical part feels underdeveloped relative to the claims being made.

A large part of the argument relies on interpreting the router as estimating something like ($P(Z \mid X)$), and then attributing the failure to mismatch between this quantity at train vs test time. This is a clean story, but it is also somewhat idealized. In practice, the router is learned jointly with the experts and is not trained to recover any true latent variable. It would help to clarify whether the results actually depend on this interpretation, or if they hold more generally.

More broadly, the paper lacks formal statements of its main claims. For example, the key result that soft routing can be miscalibrated under distribution shift—even with calibrated experts—is not presented as a theorem with explicit assumptions. Instead, the argument is given at a more informal level. The same applies to the robustness of hard routing.

Similarly, the DRO formulation is well-motivated, but the paper does not provide generalization guarantees or a clear characterization of when the chosen uncertainty set (bounded reweighting) is appropriate. The multiaccuracy discussion is interesting, but also fairly loose and does not translate into concrete calibration guarantees without additional assumptions.

On the empirical side, evaluation is limited. Everything is done on CIFAR-10H, and it is not clear how the findings translate to larger-scale MoE systems (e.g., language models), where routing behavior is more complex.

Finally, the method introduces a tradeoff (e.g., potentially reduced sharpness due to focusing on high-loss regions), but this is not analyzed in depth.

---

> ### Author Rebuttal · Authors · 2026-03-31
>
> Thank you for the thoughtful review. We are glad that the main idea came across clearly, and that you found it compelling. We also appreciate that you find the proposed fix simple but practical.
>
> **On the latent-variable interpretation of the router**
>
> We agree that the current presentation implies that the router estimates something like $P(Z|X)$. We would like to clarify that our theory does not actually require that the router literally recovers a true latent variable; the latent variable language is more an illustrative explanation for how routing posterior shift can occur. Routing posterior shift can be described more generally: under soft routing, the aggregate prediction depends on input-dependent mixture weights, and calibration can fail under shift when those effective mixture weights no longer align with the test distribution, even if the experts themselves are calibrated on the distributions they see. We will make this presentation much more explicit in the revision.
>
> **On formal statements**
>
> The previous issue could be mitigated if the main claims were stated more formally, as you suggest. We see two statements in particular where more formal precision would help. (1) The hard-routing result can be stated as a calibration-preservation statement under reweightings that are measurable with respect to the routing partition, making clear exactly which class of shifts leaves the aggregate predictor calibrated when each expert is calibrated on its own region. (2) For soft routing, the paper can isolate the failure mechanism as a proposition showing that once the aggregate prediction depends on input-dependent mixture weights, calibration of the experts alone no longer suffices under distribution shift.
>
> We appreciate the suggestion!
>
> **On the DRO uncertainty set and the choice of $\Gamma$**
>
> In the current experiments, the tail fraction $\Gamma$ is a parameter that is selected by validation. Our intended interpretation is that $\Gamma$ controls how aggressively the objective concentrates on vulnerable regions of the training distribution: smaller tail fractions focus the optimization more tightly on the highest-loss slices, while larger values move the objective closer to standard average-case training. In that sense, $\Gamma$ should be read as a robustness-sensitivity parameter tied to how much emphasis one wants to place on the rare but calibration-critical regions. We will make this interpretation more concrete in the revision.
>
> **On sharpness and confidence**
>
> We do not expect the method to systematically reduce confidence in a uniform sense. First, the objective does not penalize confidence, it penalizes large proper loss under worst-case reweighting, so the main effect is to reduce confidence *specifically* on slices where the model is brittle under shift. In that sense, the tradeoff is not best understood as "more robustness means less confidence everywhere," but rather as a redistribution of confidence away from regions where the soft-routed aggregate is overconfident and unstable.
>
> That said, if the robustness parameter is set too aggressively, the method can indeed sacrifice overall sharpness or average-case performance in order to protect the tail. The natural control for this tradeoff is the reweighting budget $\Gamma$.
>
> In our revision, we will make this control mechanism more explicit, and we will discuss the tradeoff in terms of both slice-wise calibration gains and any movement in full-distribution sharpness or proper loss.
>
> **On shifts beyond bounded reweighting**
>
> The current objective is designed for reweighting-style shifts, so the cleanest guarantees are in that setting. For label shift, the answer is mixed: to the extent that the shift changes the prevalence of subpopulations or expert regions in a way that can be represented as a reweighting of the training distribution, the method should help, since it explicitly trains against concentration of mass on regions where the soft-routed aggregate is fragile. But label shift in full generality is not covered by the current analysis, because once the class proportions change, the effect on the aggregate predictor does not necessarily reduce to the bounded-reweighting model we study.

---

> > ### Author Rebuttal · Reviewer_zHYU · 2026-04-05
> >
> > Thank you for the thoughtful and detailed response. We appreciate the clarifications and are glad to see that you plan to make the presentation more precise and formal in the revision.
> >
> > In particular, the clarification that the latent-variable interpretation is illustrative rather than required by the theory is helpful, and we agree that framing routing posterior shift in terms of input-dependent mixture weights under soft routing provides a more general and accurate perspective. We also welcome the proposed formalization of both the hard-routing calibration preservation result and the soft-routing failure mechanism, which should significantly strengthen the theoretical clarity of the paper.
> >
> > Your discussion of the DRO uncertainty set and the role of the tail fraction as a robustness-sensitivity parameter is also helpful. Making this interpretation explicit, along with clearer guidance on its selection and its effect on the optimization objective, will improve the practical usability of the method.
> >
> > We also appreciate the nuanced discussion of the tradeoff between robustness and sharpness. The clarification that the method redistributes confidence rather than uniformly reducing it is important, and we agree that explicitly characterizing the role of the reweighting budget in controlling this tradeoff would be valuable.
> >
> > Finally, your clarification regarding the scope of distribution shifts—particularly the distinction between bounded reweighting and more general label shift—is well taken. It would be helpful to clearly delineate these assumptions and limitations in the final version.
> >
> > Overall, we find your responses constructive and believe the proposed revisions will improve both the clarity and rigor of the paper.

---

> > > ### Author Response · Authors · 2026-04-06
> > >
> > > Thank you for your follow up!
> > >
> > > In your review, you had brought up a concern that our empirical evaluation was too limited to properly understand how our findings translate to larger scale MoE systems. We have been working on significantly expanding our experiments since then. Please feel free to check on our current progress at: https://dblindanon.github.io/
> > >
> > > For these experiments, we introduce a new variant of our proposed method, Robust Filtered, which restricts the robust tail emphasis to examples where routing is actually relevant---specifically, where the mixture prediction underperforms its best expert, or where experts meaningfully disagree. The website includes an overview of this method in R.1.
> > >
> > > ---
> > > We hope the new experiments help to answer some of your previous concerns. In particular:
> > >
> > > *"On the empirical side, evaluation is limited. Everything is done on CIFAR-10H, and it is not clear how the findings translate to larger-scale MoE systems (e.g., language models), where routing behavior is more complex."*
> > >
> > > Our new experiments include results on the PACS image dataset and the CivilComments WILDS text dataset. The distribution shifts in these datasets are more realistic, as PACS is a classic example of domain/style shift, and CivilComments is a natural example of subpopulation shift. We use the language model DistilBERT for the CivilComments dataset. (We are working with limited compute, but we particularly wanted to incorporate at least one larger model, following your suggestion and the suggestion of Reviewer pWkB.) In our experiments, we see that both robust methods, Robust MoE and Robust Filtered, significantly improve calibration on all datasets for only a small drop in accuracy. In the case of CivilComments, both robust methods improve calibration at no cost to accuracy, achieving Pareto improvement.
> > >
> > > *"The method introduces a tradeoff (e.g., potentially reduced sharpness due to focusing on high-loss regions), but this is not analyzed in depth."*
> > >
> > > The new experiments help to explore this potential tradeoff more directly. In addition to evaluating ECE on hard subpopulations, we also report ECE and reliability diagrams over all examples for every dataset. These results suggest that, although optimization focuses on high-loss regions, our methods do not simply flatten predictions globally: the reliability diagrams show that the models still make high-confidence predictions on easy examples, and that those predictions remain well calibrated. Moreover, overall ECE improves in almost all settings, including all 4 test domains of PACS and CivilComments. The only exception is CIFAR10, where Robust MoE does show a tradeoff between overall ECE and Hard ECE, although this is mitigated by temperature scaling.
> > >
> > > That said, we agree that focusing on high-loss regions should be done carefully. Our proposed variant, Robust Filtered, helps address this concern by only focusing on high-loss regions when the loss is due to routing instability.
> > >
> > > ---
> > > We hope you find these additional experiments help to demonstrate that our method works across a larger range of practical settings.

---

### Decision · Program_Chairs · 2026-04-30

**Decision:**

Accept (regular)

**Comment:**

The paper identifies a compelling and important failure mode in Mixture-of-Experts calibration.  It explains why hard routing induces an information bottleneck that protects calibration under reweighting shifts, whereas soft routing does not. The work proposes a practical solution using a distributionally robust objective. The reviewers initially raised concerns on the clarity and experimental soundness, most of which have been addressed by the author(s) in their rebuttals. They should be incorporated in the revision. Overall, the paper makes valuable contributions and should be accepted.